# LFME: A Simple Framework for Learning from Multiple Experts in Domain Generalization

**Liang Chen**[1]    **Yong Zhang**[2*]    **Yibing Song**    **Zhiqiang Shen**[1]    **Lingqiao Liu**[3*]

[1] MBZUAI    [2] Meituan Inc.    [3] The University of Adelaide

{liangchen527, zhangyong201303, yibingsong.cv}@gmail.com
Zhiqing.Shen@mbzuai.ac.ae    lingqiao.liu@adelaide.edu.au

## Abstract

Domain generalization (DG) methods aim to maintain good performance in an unseen target domain by using training data from multiple source domains. While success on certain occasions are observed, enhancing the baseline across most scenarios remains challenging. This work introduces a simple yet effective framework, dubbed learning from multiple experts (LFME), that aims to make the target model an expert in all source domains to improve DG. Specifically, besides learning the target model used in inference, LFME will also train multiple experts specialized in different domains, whose output probabilities provide professional guidance by simply regularizing the logit of the target model. Delving deep into the framework, we reveal that the introduced logit regularization term implicitly provides effects of enabling the target model to harness more information, and mining hard samples from the experts during training. Extensive experiments on benchmarks from different DG tasks demonstrate that LFME is consistently beneficial to the baseline and can achieve comparable performance to existing arts. Code is available at https://github.com/liangchen527/LFME.

## 1   Introduction

Deep networks trained with sufficient labeled data are expected to perform well when the training and test domains with similar distributions [74, 4]. However, test domains in real-world often exhibit unexpected characteristics, leading to significant performance degradation for the trained model. Such a problem is referred to as distribution shift and is ubiquitous in common tasks such as image classification [43, 27] and semantic segmentation [19, 40]. Various domain generalization (DG) approaches have been proposed to address the distribution shift problem lately, such as invariant representation learning [53, 69, 57, 62, 29], augmentation [91, 48, 79, 47], adversarial learning [24, 46, 81, 49], meta-learning [44, 2, 22, 45], to name a few. Yet, according to [27], most arts perform inferior to the classical Empirical Risk Minimization (ERM) when applied with restricted hyperparameter search and evaluation protocol. Both the experiments in [27] and our findings suggest that existing models are incapable of consistently improving ERM in all evaluated datasets. The consistent improvement for ERM thus becomes our motivation to further explore DG.

Our approach derives from the observation in [10] that some of the data encountered at test time are similar to one or more source domains, and in which case, utilizing expert models specialized in the domains might aid the model in making a better prediction. The observation can be better interpreted with the example in Fig. 1. Given experts trained in the "infograph", "real", and "quickdraw" domains, and test samples from the novel "sketch" domain. Due to the large domain shift, it would be better to rely mostly on the expert trained on the similar "quickdraw" domain than others.

---

*Corresponding authors.

38th Conference on Neural Information Processing Systems (NeurIPS 2024).

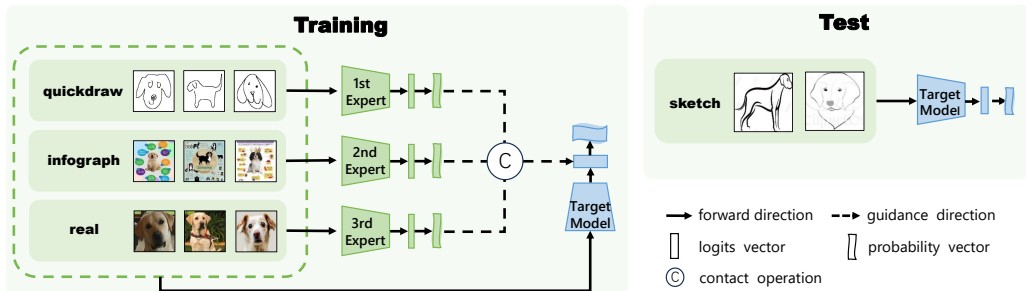

Figure 1: Pipeline of LFME. Experts and the target model are trained simultaneously. To obtain a target model that is an expert on all source domains, we learn multiple experts specialized in corresponding domains to help guide the target model during training. For each sample, the guidance is implemented with a logit regularization term that enforces similarity between the logit of the target model and probability from the corresponding expert. Only the target model is utilized in inference. Please refer to Algorithm 1 for detailed implementations.

However, the test domain information is often unavailable in DG, indicating that we can not specifically train an expert specialized in a particular domain. In light of this, obtaining a target model that is an expert on all source domains seems to be a practical alternative for handling potential arbitrary test domains. A naive implementation would be training multiple experts on each domain, and dynamically aggregate them to form the target model. So that any encountered test samples can be predicted by corresponding experts who are familiar with their characteristics. Nevertheless, such a practice has two inherent limitations: (1) designing an effective aggregation mechanism is essential and inevitable for the naive model. In fact, our experimental study indicates that using naive aggregating strategies, such as averaging, may deteriorate the performance. (2) the overall framework requires much more resources for deployment when there are many training domains, since all the experts must be leveraged during the test phase.

This work proposes a simple framework for learning from multiple experts (LFME), capable of obtaining an expert specialized in all source domains while avoiding the aforementioned limitations. Specifically, during the training stage, instead of heuristically aggregating different experts, we suggest training a universal target model that directly inherits knowledge from all these experts, which is achieved by a simple logit regularization term that enforces the logit of the target model to be similar to probability from the corresponding expert. With this approach, the target model is expected to leverage professional guidance from multiple experts, evolving into an expert across all source domains. During the test phase, only the target model is utilized. As a result, both model aggregations and extra memory and computation resources are not required during the deployment, since we only leverage one model. The overall training and test pipelines are illustrated in Fig. 1.

Our method can be interpreted through the lens of knowledge distillation (KD), where the core idea is transferring knowledge by training the student (*i.e.* the target model) with soft labels from teachers (*i.e.* experts) [31]. Unlike traditional KD [31] that uses teachers' output probabilities as soft labels in a cross entropy loss, we employ a logit regularization term that uses experts' probabilities to refine the logit of the target model in a regression manner, which can be regarded as extending the effectiveness of mean squared error (MSE) loss in classification [37] within the KD realm.

To gain a deeper understanding of the effectiveness of our logit regularization term, we perform in-depth analyses and uncover that its merit over the baseline can be explained in twofold. (1) It implicitly regularizes the probability of the target model within a smaller range, enabling it to use more information for prediction and improve DG accordingly. It is noteworthy that the effect is achieved inherently differs from that by label smoothing (LS) [71], as LFME does not require explicitly calibration for the output probability. Expanding upon this analysis, we find that a simple combination of cross entropy and MSE losses achieves comparable performance among existing arts. Given its straightforward implementation without unnecessary complexities, this expanding may offer a "free lunch" for DG; (2) It further boosts generalization by helping the target model to focus more on hard samples from the experts, supported by our theoretical finding. Through experiments on different datasets, we find that hard samples from the experts are more beneficial for generalization than those from the model itself. Given that hard sample mining is essential for easing distribution shift [35, 42], this discovery may offer valuable guidance for future research endeavors.

Through evaluations on the classification task with the DomainBed benchmark [27] and segmentation task with the synthetic [63, 64] to real [20, 83, 55] setting, we illustrate that LFME is consistently beneficial to the baseline and can obtain favorable performance against current arts (other KD ideas included). *Our method favors extreme simplicity, adding only one hyper-parameter, that can be tuned in a rather large range, upon training the baseline ERM.*

## 2    Related Works

**General DG methods.** Domain generalization (DG), designed to enable a learned model to maintain robust results in unknown distribution shift, is gaining increasing attention in the research community lately. The problem can be traced back to a decade ago [6], and various approaches and applications have been proposed to push the generalization boundary ever since [53, 26, 46, 32, 81, 49, 2, 16, 11, 13, 82, 12]. The pioneering work [4] theoretically proves that the DG performance is bounded by both the intra-domain accuracy and inter-domain differences. Most previous arts focus on reducing the inter-domain differences by learning domain-invariant features with ideas such as kernel methods [53, 26], feature disentanglement [10, 61], and gradient regularization techniques [69, 62]. Same endeavors also include different learning skills: adversarial training is leveraged [24, 81] to enforce representations to be domain agnostic; meta-learning is utilized [2, 45] to simulate distribution shifts during training. Other works aim to improve the intra-domain accuracy: some suggest explicitly mining hard samples or representations with handcraft designs, such as masking out dominant features [36], weighting more on hard samples [42], or both [35]. LFME falls into this category as the target model can also mine hard samples from the experts, beneficial for excelling in all source domains (in Sec. 6.5).

**Utilizing experts for DG.** Methods with the most relevant motivations with our LFME are perhaps those also involves experts [86, 28, 89, 90]. In DAELDG [90], a shared feature extractor is adopted, which is followed by different classifiers (*i.e.* expert) that correspond to specific domains. Their experts are trained by enforcing the outputs to be similar to the average output from the non-expert classifiers. Different from our work, it uses the average outputs from different experts as the final result. In Meta-DMoE [89], similar to LFME, different experts are trained on their specific domains where a traditional KD idea is adopted: the feature from the target model is enforced to be similar to the transformer-processed version of their experts' output features. Notably, Meta-DMoE and LFME share very distinct objectives for the expert models. Specifically, Meta-DMoE aims to adapt the trained target model to a new domain in test. To facilitate adaptation, their target model is assumed to be capable of identifying domain-specific information (DSI), and is enforced to extract DSI similar to those from domain experts. In their settings, domain experts are expected to thrive in all domains and are used not in their trained domains but rather in an unseen one. Differently, LFME expects its target model to be expert in all source domains, where domain experts provide professional guidance for the target model only in their corresponding domains. Additionally, Meta-DMoE involves meta-learning and test-time training, which is more complicated than the end-to-end training adopted in LFME. Unlike their empirical design, our method is more self-contained, supported by in-depth analysis to explain its efficacy (in Sec. 4). We further show in our experiments (in Sec. 5.1 and E) that these related two methods perform inferior to our design, and the improvements from their basic designs: using average performance (in Sec. 6.4) or enforcing feature similarity between the target model and experts (in Sec. 6.3) are subtle compared to our logit regularization.

**DG in semantic segmentation.** Different from image classification, semantic segmentation involves classifying each pixel of the image, and the generalizing task often expects a model trained from the synthetic environments to perform well on real-world views. Directly extending general DG ideas to semantic segmentation is not easy. Current solutions mainly consist of domain randomization [34, 85], normalization [19, 56, 73], or using some task-related designs, such as the class memory bank in [40]. Different from some existing DG methods, LFME can be directly extended to ease the distribution shift problem in the semantic segmentation task without requiring any tweaks, and we show that it can obtain competitive performance against recent arts specially designed for the task.

## 3    Methodology

**Problem Setting.** In the vanilla DG setting, we are given $M$ source domains $\mathcal{D}_s = \{\mathcal{D}_1, \mathcal{D}_2, ..., \mathcal{D}_M\}$, where $\mathcal{D}_i$ is the $i$-th source domain containing data-label pairs $(x_n^i, y_n^i)$ sampled from different

probabilities on the joint space $\mathcal{X} \times \mathcal{Y}$, the goal is to learn a model from $\mathcal{D}_s$ for making predictions on the data from the unseen target domain $\mathcal{D}_{M+1}$. For either DG or the downstream semantic segmentation task, source and target domains are considered with an identical label space, and we assume that there are a total of $K$ classes.

**Learning multiple experts**. LFME trains all the experts and the target model simultaneously, and the training procedure is illustrated in the upper part of Fig. 1. As each expert corresponds to a specific domain, given the $M$ source domains, a total of $M$ experts have to be trained during this stage, and the training process of each expert is the same as that of the ERM model. Given a training batch $\mathcal{B} \in \mathcal{D}_s$, for the $i$-th expert $E_i$, we only use data from the $i$-th domain, and the computed loss regarding the data-label pair $(x^i, y^i) \in (\mathcal{B} \cap \mathcal{D}_i)$ can be written as,

$$\mathcal{L}_i = \mathcal{H}(q^{E_i}, y^i), \text{ s.t. } q_c^{E_i} = \text{softmax}(z_c^{E_i}) = \frac{exp(z_c^{E_i})}{\sum_j^K exp(z_j^{E_i})}, \tag{1}$$

where $q^{E_i} \in \mathbb{R}^K$ is the output probability computed by applying the softmax function over the output logits $z^{E_i}$ with $z^{E_i} = E_i(x^i)$; $\mathcal{H}$ denotes the cross-entropy loss: $\mathcal{H}(q, y) = \sum_c^K -y_c \log q_c$.

**Learning the target model**. Data-label pairs $(x, y) \in \mathcal{B}$ from all domains are used for training the target model $T$. The main classification loss $\mathcal{L}_{cla}$ is computed similar to Eq. (1): $\mathcal{L}_{cla} = \mathcal{H}(q, y)$, s.t. $q = \text{softmax}(z) = \frac{exp(z_c)}{\sum_j^K exp(z_j)}$ and $z = T(x)$. Then, to incorporate professional guidance from the experts, we further introduce a logit regularization term $\mathcal{L}_{guid}$, to assist $T$ to become an expert on all source domains, which is computed by using the probabilities from the experts as a label for $T$:

$$\mathcal{L}_{guid} = \|z - q^E\|^2, \tag{2}$$

where $q^E$ is the concatenate probabilities from different experts along the batch dimension [2], $\|\cdot\|$ denotes the $L_2$ norm, and this term is only enforced on the target model. Note we use the normalized version of $z^E$ (*i.e.* $q^E$) for computing $\mathcal{L}_{guid}$, which can be regarded as extending the effectiveness of MSE loss [37] in the KD realm. Our experimental studies (in Sec. 6.3) find it leads to better performance, and the following contents also elaborate on the motivation. Then, the overall loss $\mathcal{L}_{all}$ for updating the target model can be represented as,

$$\mathcal{L}_{all} = \mathcal{L}_{cla} + \frac{\alpha}{2} \mathcal{L}_{guid}, \tag{3}$$

where $\alpha$ is the weight parameter, the only additional parameter upon ERM. We train the target model and the experts simultaneously for simplicity. Please refer to pseudocode in Algorithm 1 for details.

**Rationality (comprehension from a KD perspective).** Our logit regularization term can be viewed as a new KD form, wherein the fundamental principle is to utilize the teachers' (*i.e.* experts) outputs as soft labels for the student (*i.e.* target model) in a training objective [31]. In the context of classification tasks, the cross entropy loss $\mathcal{H}(q, y)$ is widely used in the literature. Based on this objective, an intuitive revision to achieve distillation is by replacing the ground-truth label $y$ with $q^E$ in a cross entropy regularization manner (*i.e.* $\mathcal{H}(q, q^E)$), which builds the rationality for the pioneering KD art [31]. Nevertheless, a recent study [37] suggests that the MSE loss $\|z - y\|$ (without applying softmax function on $z$) performs as well as the cross entropy loss when being applied in the classification task. Correspondingly, a distillation scheme motivated by this objective can thus be utilizing the soft label $q^E$ in a regression manner, which comes to our logit-regularized term: $\|z - q^E\|$. From this perspective, the rationality of the introduced logit regularization term aligns with the principle of KD, and it can be regarded as extending the applicability of MSE loss in classification to the KD realm. We compare our new KD form with other ideas in Sec 6.3, demonstrating its superior performance against existing KD ideas in the DG task. We delve deep into our method and explain the effectiveness of LFME in the following section.

**Computational cost.** Inherited from KD, training LFME inevitably requires more resources as both the teacher and student have to be involved during training. However, LFME uses the same test resources as the baseline ERM given only the target model is utilized. Meanwhile, please also note that the computational cost for LFME is not proportional w.r.t the domain size. Instead, the training cost will always be doubled compared to ERM, as each sample will require two forward passes: one for the target model and the other for the corresponding expert. Please refer to Tab. 11 for training time comparisons between different arts. In Sec. F, we show that simply increasing training resources for current arts cannot improve their performances, suggesting it may not be a primary factor in DG.

---

[2]$z$ can also be regarded as concatenated results from different domains: $\mathcal{L}_{guid} = \sum_i \mathcal{L}_{guid}^i = \sum_i \|z^i - q^{E_i}\|^2$.

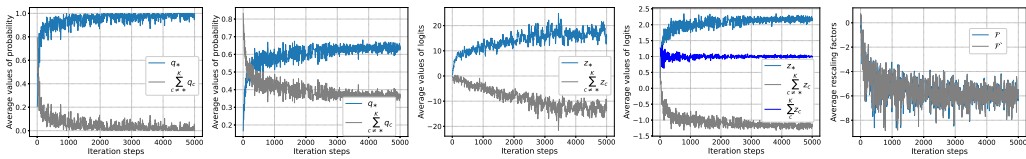

| (a) $q$ from ERM | (b) $q$ from LFME | (c) $z$ from ERM | (d) $z$ from LFME | (e) $\mathcal{F}$ and $\mathcal{F}'$ |

Figure 2: Values of probabilities, logits, and rescaling factors(*i.e.* $q$, $z$, $\mathcal{F}$, $\mathcal{F}'$) from the ERM model and LFME. Models are trained on three source domains from PACS with the same settings.

# 4 Deeper Analysis: How the Simple Logit Regularization Term Benefits DG?

## 4.1 Enabling the Target Model to Use More Information

Specifically, for the baseline model, using only the classification loss $\mathcal{L}_{cla}$ encourages the probability $q$ to be diverse: the ground truth $q_*$ to approximate 1 and 0 for others. Consequently, the corresponding logits $z_*$ will increase ceaselessly, *i.e.* $z_* \rightarrow +\infty$, and vice versa for $z_c$, *i.e.* $z_c \rightarrow -\infty$, $\forall c \neq *$ (depicted in Fig. 2 (a) and (c)), as this is the solution for minimizing $-\log \frac{exp(z_*)}{\sum exp(z_c)}$. From another point of view, $\mathcal{L}_{cla}$ encourages the model to explicitly focus on the dominant and exclusive features that are strongly discriminative but may be biased towards simplistic patterns [25].

Differently, when $\mathcal{L}_{guid}$ is imposed, the logits $z$ will approximate the range of $q^E$ (*i.e.* $[0, 1]$). Eventually, the final logits will balance these two losses (*i.e.* $z_* \in (q_*^E, +\infty)$ and $z_c \in (-\infty, q_c^E)\forall c \neq *$ as shown in Fig. 2 (d)), resulting in a smoother distribution of $q$, where $q_c$ ($\forall c \neq *$) in LFME will be larger than it is in ERM (see Fig. 2 (b)). Since both two losses encourage the model to make a good prediction (*i.e.* $z_*$ is expected to be the largest in both losses), the increase of $q_c$ indicates that besides learning the dominant features, the target model will be enforced to learn other information that is shared with others. Compared with ERM that prevents the model from learning other features, LFME is more likely to make good predictions when certain types of features are missing while others exist in unseen domains. We provide a "free lunch" inspired by the analysis in Sec. 6.1, and more justifications (including visual and empirical evidence) to support this analysis in Sec. D.1 and D.2. In Sec. C, we demonstrate that other KD ideas face challenges in achieving the same merit.

The above finding can also be endorsed by the information theory [68]. Specifically, with a smoother distribution of $q$, the entropy, which measures information, will naturally increases [51], suggesting a theoretical basis for utilizing more information in LFME. According to the principle of maximum entropy [38], the improvement for generalization is thus foreseeable [88].

Note this effect is achieved inherently differs from that by label smoothing (LS) [71], as LFME does not involve hand-crafted settings to deliberately calibrate the output probability, which is essential in LS. In Sec. 6.2, we show LS is ineffective in DG compared to LFME. Besides the advantage of avoiding problems raised by potential improper heuristic designs, we show in the following that the logit regularization in LFME provides another merit over LS.

## 4.2 Enabling the Target Model to Mine Hard Samples from the Experts

This effect is realized by example re-weighting, motivated by proposition 2 in [72]. The single sample gradient of $\mathcal{L}_{all}$ with respect to the $c$-th logit value $z_c$ can be formulated as,

$$\frac{\partial \mathcal{L}_{all}}{\partial z_c} = q_c - y_c + \alpha(z_c - q_c^E). \tag{4}$$

When the target probability corresponds to the ground-truth $y_* = 1$, Eq. (4) is reduced into,

$$\frac{\partial \mathcal{L}_{all}}{\partial z_*} = q_* - 1 + \alpha(z_* - q_*^E). \tag{5}$$

In this situation, the rescaling factor $\mathcal{F}$ is given by,

$$\mathcal{F} = \frac{\partial \mathcal{L}_{all}}{\partial z_*} / \frac{\partial \mathcal{L}_{cla}}{\partial z_*} = 1 - \alpha \frac{z_* - q_*^E}{1 - q_*}. \tag{6}$$

On the other hand, for $\forall c \neq *$, summing the gradient values over the finite indexes gives,

$$\sum_{c \neq *} \frac{\partial \mathcal{L}_{all}}{\partial z_c} = \sum_{c \neq *} q_c + \alpha \sum_{c \neq *}(z_c - q_c^E), \tag{7}$$

Table 1: Evaluations in DomainBed with default settings (3 random seeds each with 20 trials). Top5 and score count how often a method achieves the top 5 performance and outperforms ERM. Results with † are from the ResNet50 backbone and others are with ResNet18. Best and second bests results are highlighted. Results with SWAD are cited from [8], and all others are reevaluated in our device.

| | PACS | VLCS | OfficeHome | TerraInc | DomainNet | Avg. | Top5↑ | Score↑ |
|---|---|---|---|---|---|---|---|---|
| MMD [46] | 81.3 ± 0.8 | 74.9 ± 0.5 | 59.9 ± 0.4 | 42.0 ± 1.0 | 7.9 ± 6.2 | 53.2 | 0 | 2 |
| RSC [36] | 80.5 ± 0.2 | 75.4 ± 0.3 | 58.4 ± 0.6 | 39.4 ± 1.3 | 27.9 ± 2.0 | 56.3 | 0 | 1 |
| IRM [1] | 80.9 ± 0.5 | 75.1 ± 0.1 | 58.0 ± 0.1 | 38.4 ± 0.9 | 30.4 ± 1.0 | 56.6 | 0 | 1 |
| DANN [24] | 79.2 ± 0.3 | 76.3 ± 0.2 | 59.5 ± 0.5 | 37.9 ± 0.9 | 31.5 ± 0.1 | 56.9 | 0 | 1 |
| GroupGRO [66] | 80.7 ± 0.4 | 75.4 ± 1.0 | 60.6 ± 0.3 | 41.5 ± 2.0 | 27.5 ± 0.1 | 57.1 | 0 | 1 |
| VREx [42] | 80.2 ± 0.5 | 75.3 ± 0.6 | 59.5 ± 0.1 | 43.2 ± 0.3 | 28.1 ± 1.0 | 57.3 | 1 | 1 |
| CAD [65] | 81.9 ± 0.3 | 75.2 ± 0.6 | 60.5 ± 0.3 | 40.5 ± 0.4 | 31.0 ± 0.8 | 57.8 | 0 | 2 |
| CondCAD [65] | 80.8 ± 0.5 | 76.1 ± 0.3 | 61.0 ± 0.4 | 39.7 ± 0.4 | 31.9 ± 0.7 | 57.9 | 0 | 1 |
| MTL [5] | 80.1 ± 0.8 | 75.2 ± 0.3 | 59.9 ± 0.5 | 40.4 ± 1.0 | 35.0 ± 0.0 | 58.1 | 0 | 0 |
| ERM [75] | 79.8 ± 0.4 | 75.8 ± 0.4 | 60.6 ± 0.2 | 38.8 ± 1.0 | 35.3 ± 0.1 | 58.1 | 0 | - |
| MixStyle [91] | 82.6 ± 0.4 | 75.2 ± 0.7 | 59.6 ± 0.8 | 40.9 ± 1.1 | 33.9 ± 0.1 | 58.4 | 1 | 1 |
| MLDG [44] | 81.3 ± 0.2 | 75.2 ± 0.3 | 60.9 ± 0.2 | 40.1 ± 0.9 | 35.4 ± 0.0 | 58.6 | 0 | 1 |
| Mixup [80] | 79.2 ± 0.9 | 76.2 ± 0.3 | 61.7 ± 0.5 | 42.1 ± 0.7 | 34.0 ± 0.0 | 58.6 | 0 | 2 |
| MIRO [9] | 75.9 ± 1.4 | 76.4 ± 0.4 | 64.1 ± 0.4 | 41.3 ± 0.2 | 36.1 ± 0.1 | 58.8 | 3 | 3 |
| Fishr [62] | 81.3 ± 0.3 | 76.2 ± 0.3 | 60.9 ± 0.3 | 42.6 ± 1.0 | 34.2 ± 0.3 | 59.0 | 0 | 2 |
| Meta-DMoE [89] | 81.0 ± 0.3 | 76.0 ± 0.6 | 62.2 ± 0.1 | 40.0 ± 1.2 | 36.0 ± 0.2 | 59.0 | 1 | 3 |
| SagNet [54] | 81.7 ± 0.6 | 75.4 ± 0.8 | 62.5 ± 0.3 | 40.6 ± 1.5 | 35.3 ± 0.1 | 59.1 | 1 | 2 |
| SelfReg [39] | 81.8 ± 0.3 | 76.4 ± 0.7 | 62.4 ± 0.1 | 41.3 ± 0.3 | 34.7 ± 0.2 | 59.3 | 1 | 3 |
| Fish [69] | 82.0 ± 0.3 | 76.9 ± 0.2 | 62.0 ± 0.6 | 40.2 ± 0.6 | 35.5 ± 0.0 | 59.3 | 1 | 4 |
| CORAL [70] | 81.7 ± 0.0 | 75.5 ± 0.4 | 62.4 ± 0.4 | 41.4 ± 1.8 | 36.1 ± 0.2 | 59.4 | 1 | 3 |
| SD [60] | 81.9 ± 0.3 | 75.5 ± 0.4 | 62.9 ± 0.2 | 42.0 ± 1.0 | 36.3 ± 0.2 | 59.7 | 2 | 4 |
| CausEB [17] | 82.4 ± 0.4 | 76.5 ± 0.4 | 62.2 ± 0.1 | 43.2 ± 1.3 | 34.9 ± 0.1 | 59.8 | 3 | 4 |
| ITTA [15] | 83.8 ± 0.3 | 76.9 ± 0.6 | 62.0 ± 0.2 | 43.2 ± 0.5 | 34.9 ± 0.1 | 60.2 | 3 | 4 |
| RIDG [14] | 82.8 ± 0.3 | 75.9 ± 0.3 | 63.3 ± 0.1 | 43.7 ± 0.5 | 36.0 ± 0.2 | 60.3 | 4 | 4 |
| Ours | 82.4 ± 0.1 | 76.2 ± 0.1 | 63.2 ± 0.1 | 46.3 ± 0.5 | 36.1 ± 0.1 | 60.8 | 4 | 5 |
| ERM† [75] | 83.1 ± 0.9 | 77.7 ± 0.8 | 65.8 ± 0.3 | 46.5 ± 0.9 | 40.8 ± 0.2 | 62.8 | - | - |
| Fish† [69] | 84.0 ± 0.3 | 78.6 ± 0.1 | 67.9 ± 0.5 | 46.6 ± 0.4 | 40.6 ± 0.2 | 63.5 | - | - |
| CORAL† [70] | 85.0 ± 0.4 | 77.9 ± 0.2 | 68.8 ± 0.3 | 46.1 ± 1.2 | 41.4 ± 0.0 | 63.9 | - | - |
| SD† [60] | 84.4 ± 0.2 | 77.6 ± 0.4 | 68.9 ± 0.2 | 46.4 ± 2.0 | 42.0 ± 0.2 | 63.9 | - | - |
| Ours† | 85.0 ± 0.5 | 78.4 ± 0.2 | 69.1 ± 0.3 | 48.3 ± 0.9 | 42.1 ± 0.1 | 64.6 | - | - |
| ERM w/ SWAD† | 88.1 ± 0.1 | 79.1 ± 0.1 | 70.6 ± 0.2 | 50.0 ± 0.3 | 46.5 ± 0.1 | 66.9 | - | - |
| CORAL w/ SWAD† | 88.3 ± 0.1 | 78.9 ± 0.1 | 71.3 ± 0.1 | 51.0 ± 0.1 | 46.8 ± 0.0 | 67.3 | - | - |
| Ours w/ SWAD† | 88.7 ± 0.2 | 79.7 ± 0.1 | 73.1 ± 0.2 | 53.4 ± 0.4 | 47.5 ± 0.0 | 68.5 | - | - |

and the rescaling factor $\mathcal{F}'$ in this situation is,

$$\mathcal{F}' = \sum_{c \neq *} \frac{\partial \mathcal{L}_{all}}{\partial z_c} / \sum_{c \neq *} \frac{\partial \mathcal{L}_{cla}}{\partial z_c} = 1 - \alpha \frac{1 - \sum_{c \neq *} z_c - q_*^E}{1 - q_*}. \quad (8)$$

We observe that both $\mathcal{F}$ and $\mathcal{F}'$ are strictly monotonically increased regarding the value of $q_*^E$. Given almost all values of the rescaling factors are negative as can be observed in Fig. 2 (e) (except in the few initial steps where $z_*$ and $q_*^E$ are both small and $\mathcal{L}_{guid}$ barely contributes) [3], with the same logits, a smaller $q_*^E$, in which case the expert is less confident, will lead to larger $|\mathcal{F}|$ and $|\mathcal{F}'|$. This phenomenon indicates that with the logit regularization term, the target model will focus more on the harder samples from the experts. Empirical findings supporting this analysis are in Sec. D.3 and D.4. Note that the segmentation task also utilizes one-hot labels and cross-entropy loss, making it applicable to the two analyses presented.

## 5 Experiments

### 5.1 Generalization in Image classification

**Datasets and Implementation details.** We conduct experiments on 5 datasets in DomainBed [27], namely PACS [43] (9,991 images, 7 classes, 4 domains), VLCS [23] (10,729 images, 5 classes, 4

---

[3] We also observe plots of $\mathcal{F}$ and $\mathcal{F}'$ almost overlap in Fig. 2 (e). This is because optimizing $\|z - q_*^E\|^2$ leads to $\sum_c (z_c - q_c^E) = 0$, enforcing $\sum_c z_c \approx q_*^E = 1$. Since there is no other constraint for $\sum_c z_c$ ($\mathcal{L}_{cla}$ does not impose a constraint on the summation of logits), we will have $\sum_c z_c \approx 1$ and $\mathcal{F} \approx \mathcal{F}'$ in all cases, this is consistent with the observation in Fig. 2 (d).

Table 2: Evaluations on the semantic segmentation task. Results with $^\dagger$ are directly cited from [19], others are reevaluated in our device. Best results are colored as red.

| | Cityscapes (%) | | BDD100K (%) | | Mapillary (%) | | Avg. (%) | |
|---|---|---|---|---|---|---|---|---|
| | mIOU | mAcc | mIOU | mAcc | mIOU | mAcc | mIOU | mAcc |
| Baseline$^\dagger$ | 35.46 | - | 25.09 | - | 31.94 | - | 30.83 | - |
| IBN-Net [56]$^\dagger$ | 35.55 | - | 32.18 | - | 38.09 | - | 35.27 | - |
| RobustNet [19]$^\dagger$ | 37.69 | - | 34.09 | - | 38.49 | - | 36.76 | - |
| Baseline | 37.19 | 48.75 | 27.95 | 39.04 | 32.01 | 48.88 | 32.38 | 45.56 |
| PinMem [40] | 41.86 | 48.30 | 34.94 | 44.11 | 39.41 | 49.87 | 38.74 | 47.43 |
| SD [60] | 34.77 | 46.63 | 28.00 | 40.33 | 31.41 | 48.18 | 31.39 | 45.05 |
| Ours | 38.38 | 48.99 | 35.70 | 46.16 | 41.04 | 53.71 | 38.37 | 49.62 |

domains), OfficeHome [76] (15,588 images, 65 classes, 4 domains), TerraInc [3] (24,788 images, 10 classes, 4 domains), and DomainNet [59] (586,575 images, 345 classes, 6 domains). We use the ImageNet [21] pretrained ResNet [30] as the backbone for both the experts and the target model. Following the designs in DomainBed, the hyper-parameter $\frac{\alpha}{2}$ in Eq. (3) is randomly selected in a range of $[0.01, 10]$. To ensure fair comparisons, all methods are reevaluated using the default settings in DomainBed in the same device with each of them evaluated for $3 \times 20$ times in different domains. Training-domain validation is adopted as the evaluation protocol. Other settings (batch size, learning rate, dropout, etc.) are dynamically selected for each trial according to DomainBed.

**Results with ResNet18.** Results are listed in Tab. 1. We observe that ERM can obtain competitive results among the models compared, which leads more than half of the sophisticated designs, and only 6 methods lead ERM in most datasets (with scores $\geq 3$). We also note that current strategies cannot guarantee improvements for ERM in all situations, given that none of them with a score of 5. Differently, our method can consistently improve ERM in all evaluated datasets and lead others in average accuracy. Specifically, our approach obtains the leading results in 4 out of the 5 datasets, and it also improves ERM by a large margin (nearly 8pp) in the difficult TerraInc dataset. Compared to methods that explicitly explore hard samples or representations (*i.e.* VREx and RSC) and that use MoE (*i.e.* Meta-DMoE), the performances of LFME are superior to them in all cases.

**Results with ResNet50.** Because larger networks require more training resources, we only reevaluate some of leading methods (*i.e.* ERM, Fish, CORAL, and SD). in our device when experimenting with ResNet50. We note that our method surpasses the baseline ERM model in all datasets and leads it by 1.8 in average. Meanwhile, our method can still outperform the second best (*i.e.* SD) by 0.7 in average. These results indicate that our method can consistently improve the baseline model, and it can perform favorably against existing arts when implemented with a larger ResNet50 backbone.

We also combine LFME with SWAD [8]. Same with the original design, the hyper-parameter searching space in this setting is smaller than the original DomainBed. We use the reported numbers in [8] for comparisons. As shown, our method can also improve the baseline and obtain competitive results when combined with SWAD. These results further validate the effectiveness of our method.

## 5.2 Generalization in Semantic Segmentation

**Datasets and Implementation details.** The training and test processes of the compared algorithms involve 5 different datasets: 2 synthetics for training, where each dataset is considered a specific domain, and 3 real datasets for evaluation. Synthetic: GTAV [63] has 24,966 images from 19 categories; Synthia [64] contains 9,400 images of 16 categories. Real: Cityscapes [20] 3,450 finely annotated and 20,000 coarsely-annotated images collected from 50 cities; Bdd100K [83] contains 8K diverse urban driving scene images; Mapillary [55] includes 25K street view images. Following the design [40], we use the pretrained DeepLabv3+ [18] and ResNet50 as the segmentation backbone for experiments. The maximum iteration step is set to 120K with a batch size of 4, and the evaluations are conducted after the last iteration step. The hyper-parameter $\frac{\alpha}{2}$ in Eq. (3) is fixed as 1 in this experiment for simplicity. We use the mean Intersection over Union (mIoU) and mean accuracy (mAcc) averaged over all classes as criteria to measure the segmentation performance.

**Experimental results.** Results are shown in Tab. 2. To better justify the effectiveness of our method, we reevaluate the baseline model, which aggregates and trains on all source data, and PinMem in our device. We also implement SD [60], which is a leading method in the image classification

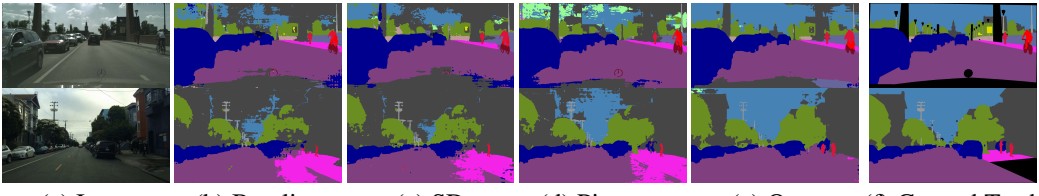

| (a) Input | (b) Bsseline | (c) SD | (d) Pinmem | (e) Ours | (f) Ground Truth |

Figure 3: Qualitative comparisons. The compared methods make unsatisfactory predictions about several objects, such as clouds with varying shapes, car logo, or people and car in the shadow. Please zoom in for a better view.

Table 3: Evaluations of free lunch for DG (*i.e.* ERM+) and different LS ideas (*i.e.* LS [71], MbLS [50], and ACLS [58]). We use the suggested settings in their original papers for evaluating.

| Model | Target domain in PACS | | | | Avg. | Target domain in TerraInc | | | | Avg. |
|---|---|---|---|---|---|---|---|---|---|---|
| | Art | Cartoon | Photo | Sketch | | L100 | L38 | L43 | L46 | |
| ERM | 78.0 ± 1.3 | 73.4 ± 0.8 | 94.1 ± 0.4 | 73.6 ± 2.2 | 79.8 ± 0.4 | 42.1 ± 2.5 | 30.1 ± 1.2 | 48.9 ± 0.6 | 34.0 ± 1.1 | 38.8 ± 1.0 |
| ERM+ | 81.9 ± 0.4 | 75.1 ± 0.7 | 94.8 ± 0.7 | 73.8 ± 2.2 | 81.4 ± 0.5 | 46.7 ± 2.6 | 37.1 ± 1.3 | 53.2 ± 0.4 | 34.8 ± 1.3 | 42.9 ± 0.7 |
| LS | 81.0 ± 0.3 | 75.4 ± 0.6 | 94.9 ± 0.2 | 73.3 ± 1.0 | 81.2 ± 0.2 | 48.1 ± 2.8 | 33.0 ± 1.6 | 53.0 ± 0.6 | 34.1 ± 1.6 | 42.1 ± 0.5 |
| MbLS | 81.3 ± 0.5 | 75.2 ± 0.6 | 94.8 ± 0.4 | 75.6 ± 0.8 | 81.7 ± 0.2 | 44.9 ± 3.3 | 39.1 ± 2.3 | 52.2 ± 0.9 | 33.8 ± 1.1 | 42.5 ± 1.5 |
| ACLS | 80.8 ± 0.3 | 75.9 ± 1.0 | 94.9 ± 0.4 | 72.3 ± 3.9 | 81.0 ± 0.6 | 45.6 ± 4.7 | 36.8 ± 0.8 | 48.9 ± 1.3 | 34.1 ± 1.7 | 41.4 ± 1.4 |
| LFME | 81.0 ± 0.9 | 76.5 ± 0.9 | 94.6 ± 0.5 | 77.4 ± 0.2 | 82.4 ± 0.1 | 53.4 ± 0.4 | 40.7 ± 2.4 | 54.9 ± 0.4 | 36.4 ± 0.7 | 46.3 ± 0.5 |

task. Results from other methods are directly cited from [19]. We observe SD does not perform as effectively as it does in the classification task, which decreases the baseline model in most situations. In comparison, our LFME can boost the baseline in all datasets, similar to that in the classification task. It also performs favorably against existing methods specially designed for the semantic segmentation task, obtaining best results in 2 out of the 3 evaluated datasets in both mIOU and mAcc.

Visualized examples are provided in Fig. 3. We note that when it comes to unseen objects (*i.e.* clouds with different shapes and a new car logo), or objects with an unfamiliar background, such as the person and car hidden in the shadow, due to the large distribution shift between real and synthetic data, compared methods make unsatisfactory predictions. In comparison, LFME can provide reasonable predictions in these objects, demonstrating its effectiveness against current arts regarding generalization to new scenes. These results validate the effectiveness of LFME and its strong applicability in the generalizable semantic segmentation task.

## 6 Analysis

Analyses in this section are conducted on the widely-used PACS dataset unless otherwise mentioned. Experimental settings are same as that detailed in Sec. 5.1. Please see the appendix for more analysis.

### 6.1 A Free Lunch for DG

As stated in Sec. 4.1, when the basis of discrimination is not compromised ($q_*$ corresponds to a larger value in the label and vice versa for $q_c$ for $\forall c \neq *$), the increase of $q_c$ can encourage the model to learn more information that is shared with other classes, and use them to improve generalization. Based on this analysis, it seems that using the one-hot label to regularize the logit is also reasonable. To validate this hypothesis, we replace $q^E$ in Eq. (3) with the ground truth and reformulate $\mathcal{L}_{all}$ into,

$$\mathcal{L}_{all} = \mathcal{H}(\text{softmax}(z), y) + \frac{\alpha}{2}\|z - y\|^2, \quad \text{s.t. } z = T(x). \tag{9}$$

Denoting as ERM+, results listed in the second row in Tab. 3 suggest that this idea can improve ERM in all unseen domains. Because the hard sample information is absent in this strategy, we observe it performs inferior to LFME. However, this alternative does not require experts or any other special designs, *even the setting of the hyper-parameter α cannot be wrong: Eq.* (9) *will approximate the baseline either with α = 0 or α approximates +∞.* Thus, we argue this simple modification can serve as a free lunch to improve DG. More evaluations of this idea are in our appendix.

## 6.2 Compare with Label Smoothing

As detailed in Sec. 4.1, LFME will explicitly constrain the probability of the target model within a smaller range. This effect may resemble the LS technique that aims to implicitly calibrate the output probability. However, compared to LS, LFME has two advantages. **First**, LFME does not involve heuristic designs of hyper-parameters for determining its probability, which is essentially required in existing LS ideas, such as $\epsilon$ in [71] and predefined margin in [50, 58], avoiding the possibility of deteriorating the performance when not choosing them properly; **Second**, LFME can explicitly mine hard samples from the experts, further ensuring improvements for DG. We evaluate 3 LS methods in both PACS and the difficult TerraInc datasets: (1) the pioneer LS method from [71]); (2) margin-based LS (MbLS) [50] that penalizes logits deviate from the maxima; (3) adaptive and conditional LS (ACLS) that can adaptively determine the degree of smoothing for different classes. Results are illustrated in 3rd-5th columns in Tab. 3. We observe that although these LS methods can improve the baseline, they are all inferior to LFME, validating the effectiveness of LFME against LS.

## 6.3 Compare with Other Knowledge Distillation Ideas

To test the effectiveness of our logit regularization (*i.e.* $\|z - q^E\|^2$), we compare it with several other KD options, namely different combinations of the logits or probabilities from the target model and the experts, including $\|z - z^E\|^2$ (which is the basic design in Meta-DMoE [89]), $\|q - z^E\|^2$, and $\|q - q^E\|^2$. Moreover, we also compare it with a common practice in KD that uses the probability from the teacher (*i.e.* experts) as a label for the student (*i.e.* target model), which reformulates the Eq. (3) into: $\mathcal{L}_{all} = (1 - \frac{\alpha}{2})\mathcal{H}(q, y) + \frac{\alpha}{2}\mathcal{H}(q, q^E)$, and denoted as $\mathcal{H}(q, q^E)$ in Tab. 4. Following the suggestion in [41], we gradually increase $\frac{\alpha}{2}$ over the iteration steps to achieve better performance. Results listed in 2nd-5th columns in Tab. 4 show that not all KD strategies can improve DG and the logit regularization choice achieves the best result in terms of average accuracy. This can be explained by the analysis in Sec. 4.1 that our logit regularization can help using more information. Please also the explanation in Sec. C for details.

## 6.4 Compare with Naive Aggregation Ideas

To examine the effectiveness of LFME, we compare it with three different variants that employ all the experts during inference: (1) Avg_Expt, which averages the output from all experts for prediction, similar to DAELDG [90]; (2) Model soup (MS)_EXPT, which uniformly combines the weights of different experts. This is inspired by the MS idea in [78]; (3) Conf_Expt that utilizes the output from the most confident expert as the final prediction. Inspired by the finding in [77], the expert with the output of the smallest entropy value is regarded as the most confident one in a test sample; (4) Dyn_Expt, which is a learning-based approach that estimates

Table 4: Out-of-domain evaluations of different KD ideas (using different $\mathcal{L}_{guid}$ in Eq. (2)) and naive aggregations.

| Model | Target domain | | | | Avg. |
|---|---|---|---|---|---|
| | Art | Cartoon | Photo | Sketch | |
| ERM | 78.0±1.3 | 73.4±0.8 | 94.1±0.4 | 73.6±2.2 | 79.8±0.4 |
| Performances from different KD ideas | | | | | |
| $\|z - z^E\|^2$ | 77.8±0.6 | 73.2±0.7 | 94.1±0.5 | 74.3±1.1 | 79.9±0.2 |
| $\|q - z^E\|^2$ | 76.4±0.7 | 75.7±1.3 | 94.0±0.2 | 72.4±1.4 | 79.7±0.5 |
| $\|q - q^E\|^2$ | 81.3±1.3 | 74.8±1.4 | 94.1±0.5 | 74.0±2.8 | 81.1±1.1 |
| $\mathcal{H}(q, q^E)$ | 82.1±0.8 | 73.6±0.2 | 92.6±0.9 | 73.4±2.1 | 80.4±0.4 |
| Performances from naive aggregation ideas | | | | | |
| Avg_Expt | 78.4±1.3 | 65.0±1.6 | 92.4±0.3 | 71.8±0.5 | 76.9±0.1 |
| MS_Expt | 76.8±2.3 | 63.0±0.6 | 93.4±0.2 | 72.6±1.5 | 76.5±0.6 |
| Conf_Expt | 74.8±0.8 | 64.8±1.8 | 91.9±0.5 | 72.6±2.6 | 76.0±0.8 |
| Dyn_Expt | 68.4±1.8 | 65.1±1.1 | 92.3±0.5 | 68.1±1.9 | 73.5±0.3 |
| Ours | 81.0±0.9 | 76.5±0.9 | 94.6±0.5 | 77.4±0.2 | 82.4±0.1 |

the domain label of each sample and dynamically assigns corresponding weights to the experts via a weighting module. Results are listed in 6th-9th columns in Tab. 4, where both the designs of handcraft (*i.e.* Avg_Expt, MS_Expt, and Conf_Expt) and learning-based (*i.e.* Dyn_Expt) aggregation skills fail to improve the baseline model. This is because the hand-craft designs in Avg_Expt, MS_Expt, and Conf_Expt are rather unrealistic in practice as different models may contribute differently in the test phase, we thus cannot use a simple average or select the most confident expert for predicting. Meanwhile, the learning-based Dyn_Expt will inevitably introduce another generalization problem regarding the weighting module, complicating the setting. In comparison, LFME avoids the nontrivial aggregation design and can improve ERM in all source domains, further validating its effectiveness.

Table 5: In-domain evaluations of different models.

| Model | Source domain in PACS | | | | Avg. | Source domain in TerraInc | | | | Avg. |
|---|---|---|---|---|---|---|---|---|---|---|
| | Art | Cartoon | Photo | Sketch | | L100 | L38 | L43 | L46 | |
| ERM | 94.8 ± 0.1 | 96.2 ± 0.2 | 98.5 ± 0.2 | 96.0 ± 0.3 | 96.4 ± 0.2 | 95.2 ± 0.1 | 91.1 ± 0.1 | 89.4 ± 0.2 | 85.5 ± 0.2 | 90.3 ± 0.3 |
| Experts | 95.4 ± 0.1 | 96.3 ± 0.1 | 98.7 ± 0.3 | 96.4 ± 0.2 | 96.7 ± 0.1 | 95.9 ± 0.1 | 92.0 ± 0.1 | 90.3 ± 0.2 | 86.3 ± 0.1 | 91.1 ± 0.1 |
| Ours | 95.7 ± 0.2 | 96.8 ± 0.3 | 98.8 ± 0.2 | 96.4 ± 0.2 | 96.9 ± 0.1 | 96.1 ± 0.1 | 92.6 ± 0.2 | 90.7 ± 0.2 | 87.2 ± 0.2 | 91.6 ± 0.2 |

Table 6: Sensitivity analysis of LFME regarding the involved weight parameter $\frac{\alpha}{2}$ in Eq. (3). LFME degrades to ERM when $\frac{\alpha}{2} = 0$.

| Model | Target domain in PACS | | | | Avg. | Target domain in TerraInc | | | | Avg. |
|---|---|---|---|---|---|---|---|---|---|---|
| | Art | Cartoon | Photo | Sketch | | L100 | L38 | L43 | L46 | |
| $\frac{\alpha}{2} = 0$ | 78.0 ± 1.3 | 73.4 ± 0.8 | 94.1 ± 0.4 | 73.6 ± 2.2 | 79.8 ± 0.4 | 42.1 ± 2.5 | 30.1 ± 1.2 | 48.9 ± 0.6 | 34.0 ± 1.1 | 38.8 ± 1.0 |
| $\frac{\alpha}{2} = 0.01$ | 82.0 ± 0.4 | 75.0 ± 0.9 | 95.2 ± 0.1 | 75.1 ± 0.9 | 81.8 ± 0.2 | 46.9 ± 2.4 | 40.0 ± 1.2 | 51.9 ± 0.5 | 34.4 ± 0.6 | 43.3 ± 0.7 |
| $\frac{\alpha}{2} = 10$ | 81.5 ± 0.3 | 75.5 ± 0.5 | 94.5 ± 0.2 | 75.6 ± 0.7 | 81.8 ± 0.1 | 52.2 ± 0.5 | 37.5 ± 1.0 | 54.4 ± 0.5 | 36.8 ± 0.6 | 45.2 ± 0.2 |
| $\frac{\alpha}{2} = 100$ | 80.1 ± 0.8 | 74.3 ± 0.6 | 93.3 ± 0.5 | 75.7 ± 1.3 | 80.9 ± 0.4 | 49.0 ± 0.7 | 30.5 ± 0.8 | 52.9 ± 0.6 | 36.1 ± 0.6 | 42.1 ± 0.6 |
| $\frac{\alpha}{2} = 1000$ | 77.1 ± 1.0 | 72.7 ± 0.9 | 92.5 ± 0.1 | 74.7 ± 0.3 | 79.3 ± 0.1 | 46.5 ± 0.9 | 29.1 ± 1.2 | 49.6 ± 0.8 | 34.6 ± 1.0 | 39.9 ± 0.2 |

## 6.5 Does the Target Model Become an Expert on All Source Domains?

The training process of LFME aggregates all professional knowledge from the experts into one target model, aiming to make the target model an expert on all source domains. To examine if the goal has been achieved, we conduct in-domain validations for the target model and compare the performances with that of the experts and the ERM model in the PACS and TerraInc datasets. Note in these experiments, ERM and LFME are trained using the same leave-one-out strategy, and the performances are averaged over the trials on three different target domains. Results are listed in Tab. 5. We observe that ERM performs inferior to the experts in the in-domain setting. The results are not surprising. As stated earlier, when encountering data similar to the source domain, it would be better to rely mostly on the corresponding expert than the model also contaminated with other patterns. In comparison, our method obtains the best results in all source domains because it implicitly focuses more on the hard samples from the experts, which is shown to be an effective way to improve the performance in many arts [33, 84]. These results validate that the proposed strategy can extract professional knowledge from different experts, and enable the target model to become an expert in all source domains.

## 6.6 Selection of the Hyper-Parameter

Compared with the baseline ERM, our method involves only one additional hyper-parameter (*i.e.* $\frac{\alpha}{2}$ in Eq. 3) which is randomly selected in the range of $[0.01, 10]$ in DomainBed. To evaluate the sensitiveness of LFME regarding this hyper-parameter, we conduct experiments in the PACS and TerraInc datasets by tuning it in a larger range. As seen in Tab. 6, LFME can obtain relatively better performance with either $\frac{\alpha}{2} = 0.01$ or 10, and it performs on par with ERM even when $\frac{\alpha}{2} = 1000$. These results indicate that our method is insensitive w.r.t the hyper-parameter. This is mainly because a very large $\frac{\alpha}{2}$ will enforce the model to only learn from domain experts, given that $q_*^E$ is mostly aligned with the label $y$ (as seen in Fig. 2 (a)), the model will perform similarly to another form of the baseline [37] in this case.

## 7 Conclusion

This paper introduced a simple yet effective method for DG where the professional guidances of experts specialized in specific domains are leveraged. We achieve the guidance by a logit regularization term that enforces similarity between logits of the target model and probability of the corresponding expert. After training, the target model is expected to be an expert on all source domains, thus thriving in arbitrary test domains. Through deeper analysis, we reveal that the proposed strategy implicitly enables the target model to use more information for prediction and mine hard samples from the experts during training. By conducting experiments in related tasks, we show that our method is consistently beneficial to the baseline and performs favorably against existing arts.

**Acknowledgement.** This work was supported by the Centre for Augmented Reasoning, an initiative by the Department of Education, Australian Government.

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

# Appendix

In this section, we provide

1. Limitations and future works in Section A

2. Pseudocode of LFME in Section B.

3. Explanations of why existing KD ideas perform inferior to our LFME in Section C.

4. More analysis regarding LFME in Section D;

5. Compare with MoE-based methods and other related methods in Section E.

6. Existing Ideas with Same Training Resources in Section F.

7. Detailed settings of the learning-based aggregation method Dyn_Expt in Sec. 6.4 from the manuscript in Section G;

8. Detailed results of LFME in the unseen domains of the different datasets from DomainBed, and detailed results of LFME in the different categories from the semantic segmentation task in Section H.

## A   Limitations and Future Works

Although our method shows competitive results for generalization, there are certain occasions when it will encounter setbacks. First, LFME is not applicable for the setting where the domain labels are unavailable in the training data, such as those collected from the internet. Since the training of the experts requires data grouped by domain labels. Second, LFME cannot handle the situation when only one source domain is provided, preventing it from performing in a more difficult single-source generalization task. How to apply LFME to a more general setting will be our primary task in future works. Besides, as the designs and theoretical supports are built mainly for the classification task, finding a proper solution to extend them to the regression tasks is also a promising direction in potential future works.

## B   Pseudo Code of LFME

This section provides the Pytorch-style pseudocode of our method. As detailed in Alg. 1, the implementation of our method is embarrassingly simple, introducing only one hyper-parameter upon the baseline ERM.

## C   Compare with Current KD Based on the Analysis in Sec. 4.1

As observed in Sec. 6.3, some existing KD ideas, perform inferior to our LFME. Based on our analysis, we find this is because existing ideas have difficulties achieving the beneficial effect introduced in Sec. 4.1. As detailed, our logit regularization ensures using more information for the target model by implicitly regularizing the probability in a much smaller range. This is hard to achieve by $\mathcal{H}(q, q^E)$ and $\|q - q^E\|^2$, where their probability $q$ will be in a similar range as that in ERM because the soft label $q^E$ is still within the range of $[0, 1]$, enforcing similarity between $q$ and $q^E$ will not significantly change its distribution. Moreover, this effect is also hard to achieve by $\|z - z^E\|$ and $\|q - z^E\|^2$, which do not provide a specific range regularization effect for the probability.

## D   Further Analysis

This section provides more evidence to support the two analyses in Sec. 4 in the manuscript. The experiments are conducted on the widely-used PACS dataset and the difficult TerraInc dataset unless mentioned otherwise.

### D.1   More Justification for Enabling Target Model to Use More Information

For the baseline model, we infer that the cross-entropy loss alone will enforce the model to only learn discriminative features that are specific to the label category. Because there are no relevant features to

**Algorithm 1:** PyTorch-style pseudocode of LFME.

```python
# M: total domain numbers
# alpha: weight parameter
# lr and weight decay: selected hyper-parameters

# Initialization: experts and the target model
network = [None] * (M + 1) # M experts and 1 target model
params = []
for i in range(M + 1):
    network[i] = ResNet()
    params.append{"params": network[i].parameters()}
optimizer = Adam(params, lr, weight_decay)

# Training: experts and the target model
def train(minibatches):
    loss = 0

    # All samples from the M domains
    all_x = torch.cat([x for x, y in minibatches])
    all_y = torch.cat([y for x, y in minibatches])
    for i in range(M): # Training the M experts

        # training i-th expert using i-th domain data
        xi, yi = minimabtch[i][0], minibatch[i][1] # image-label pair from the i-th domain
        z_Ei = network[i](xi) # logit from the i-th expert
        # note the softmax function is encoded in crossentropy loss in pytorch
        loss += crossentropy(softmax(z_Ei),yi) # Eq. (1) in the manuscript

        # Concat probabilities from experts
        qE = softmax(z_Ei) if i==0 else torch.cat((qe, softmax(z_Ei)), 0)

    z = network[-1](all_x) # logit of target: T(x)
    # L_cla + L_guid: Eq. (3) in the manuscript
    loss += crossentropy(softmax(z), all_y) + alpha * MSELoss(z, qE.detach())

    # Updating the experts and target model
    optimizer.zero_grad()
    loss.backward()
    optimizer.step()

# Test: target model
def test(test_samples):
    result = network[-1](test_samples)
```

interpret the non-label classes in ERM, the confidence for the target class will approach the maximum during training, and vice versa for other non-label classes. Differently, when the output probability has a smoother distribution, the model will also learn information that is shared with other classes to improve the corresponding non-label probabilities. Introducing more information during training will inevitably cause the phenomenon of low confidence for the label category, as probabilities for the non-label classes will increase (their summation increases from nearly 0 in Fig. 2 (a) to 0.35 in Fig. 2 (b)). Note that low confidence does not mean low accuracy. Because the label category corresponds to the largest logit value in both terms, and both the discriminative and newly introduced information can be used to characterize the label information, exploring more information can also boost classification. This finding is further validated by the in-domain results in Tab. 5, where LFME is shown to be able to also improve the in-domain test results compared to ERM, indicating that low confidence in our case actually leads to higher accuracy.

Visual examples to support the above analysis are shown in Fig. 4. We observe that compared to the baseline ERM, our method can utilize more regions for prediction. Such as that in the first "horse" examples, our model utilizes both the head, tail, and body for prediction, while the ERM seems to focus only on the head region. These visual samples align with our analysis that compared to the baseline ERM, LFME tends to use more information for prediction.

### D.2   Other Alternatives to Enlarge $q_c$ for $\forall c \neq *$

This section provides more evidence to support the first analysis in Sec. 4 by conducting experiments on two variants that also enlarge $q_c$ for $\forall c \neq *$. (1) we test a model that directly enlarges $q_c$ for $\forall c \neq *$. Specifically, we use the probability from LFME as the label to guide the probability output of a new target model, *i.e.* $\mathcal{L}_{guid} = \|q - q^{LFME}\|^2$ with $q$ and $q^{LFME}$ from the new target and LFME models, respectively. Ideally, this modification (*i.e.* LFME_Guid) can also improve the baseline model because it also encourages the model to learn more information shared with other classes

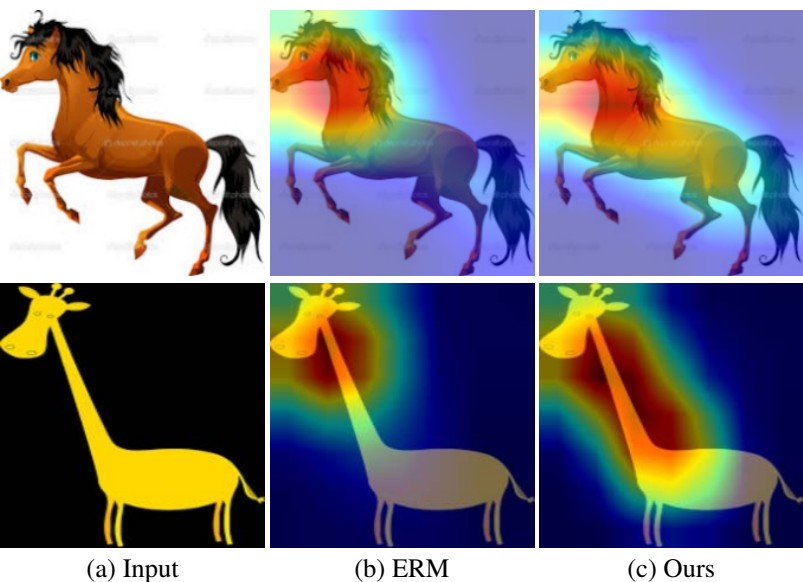

| (a) Input | (b) ERM | (c) Ours |

Figure 4: Grad-CAM visualizations of samples from the unseen "cartoon" domain of the PACS benchmark, which is the most challenging domain for ERM and our method according to Tab. 3. Compared to the baseline ERM, highlight regions from our method contain more information related to the label category. These visualizations can further validate our analysis in Sec. 4 that with the proposed strategy, the target model can explore more information for prediction.

Table 7: Further evidences to support our deep analyses in Sec. 4. Evaluations are conducted on the widely-used PACS and difficult TerraInc datasets using settings from the DomainBed benchmark. Here ERM+ is the free lunch introduced in Sec. 6.1 that replaces $q^E$ in Eq. (3) with the one-hot label; LFME_Guid denotes imposing the guidance from LFME to the ERM model by including $\mathcal{L}_{guid} = \|q - q^{LFME}\|^2$ where $q$ and $q^{LFME}$ are probabilities from the ERM model and LFME; Self_Guid is the model that replaces probabilities from the experts in Eq. (3) with that from itself, *i.e.* $\mathcal{L}_{guid} = \|z - \text{softmax}(z)\|^2$ where $\text{softmax}(z)$ is followed with a detach operation; ERM+ w/ expt denotes explicitly focus more on the hard samples from the experts based on ERM+; ERM+ w/ self denotes explicitly focus more on the hard samples from the model itself on the basis of ERM+. All methods are with the same ResNet18 backbone and are examined for 60 trials in each unseen domain.

| Model | Target domain in PACS | | | | Avg. | Target domain in TerraInc | | | | Avg. |
|---|---|---|---|---|---|---|---|---|---|---|
| | Art | Cartoon | Photo | Sketch | | L100 | L38 | L43 | L46 | |
| ERM | 78.0 ± 1.3 | 73.4 ± 0.8 | 94.1 ± 0.4 | 73.6 ± 2.2 | 79.8 ± 0.4 | 42.1 ± 2.5 | 30.1 ± 1.2 | 48.9 ± 0.6 | 34.0 ± 1.1 | 38.8 ± 1.0 |
| ERM+ | 81.9 ± 0.4 | 75.1 ± 0.7 | 94.8 ± 0.7 | 73.8 ± 2.2 | 81.4 ± 0.5 | 46.7 ± 2.6 | 37.1 ± 1.3 | 53.2 ± 0.4 | 34.8 ± 1.3 | 42.9 ± 0.7 |
| *Performances of other alternatives to enlarge $q_c$ for $\forall c \neq *$* | | | | | | | | | | |
| LFME_Guid | 81.6 ± 1.1 | 73.4 ± 0.9 | 94.9 ± 0.6 | 76.1 ± 0.4 | 81.5 ± 0.3 | 48.0 ± 2.1 | 36.8 ± 1.5 | 53.2 ± 0.8 | 35.6 ± 0.4 | 43.4 ± 0.6 |
| Self_Guid | 81.5 ± 1.0 | 75.1 ± 0.5 | 95.1 ± 0.3 | 74.0 ± 1.3 | 81.4 ± 0.4 | 48.0 ± 2.7 | 35.9 ± 0.6 | 53.9 ± 0.2 | 36.7 ± 1.9 | 43.6 ± 0.7 |
| *Comparisons between hard samples from different models* | | | | | | | | | | |
| ERM+ w/ expt | 80.8 ± 0.5 | 74.2 ± 1.1 | 95.0 ± 0.5 | 76.0 ± 0.5 | 81.5 ± 0.3 | 51.2 ± 0.5 | 37.0 ± 3.0 | 53.5 ± 0.6 | 36.5 ± 1.1 | 44.6 ± 1.0 |
| ERM+ w/ self | 82.9 ± 0.4 | 75.2 ± 0.3 | 94.0 ± 0.4 | 73.9 ± 2.2 | 81.5 ± 0.5 | 50.2 ± 0.9 | 35.7 ± 2.0 | 51.1 ± 0.9 | 34.2 ± 0.8 | 42.8 ± 0.3 |
| LFME | 81.0 ± 0.9 | 76.5 ± 0.9 | 94.6 ± 0.5 | 77.4 ± 0.2 | 82.4 ± 0.1 | 53.4 ± 0.4 | 40.7 ± 2.4 | 54.9 ± 0.4 | 36.4 ± 0.7 | 46.3 ± 0.5 |

without compromising discrimination; (2) we use the output probability from the target model itself to replace that from the experts in Eq. (3), which reformulates $\mathcal{L}_{guid}$ into $\|z - \text{softmax}(z)\|^2$ ($\text{softmax}(z)$ is followed with a detach operation in updating). According to our analyses in Sec. 4, this alternative (*i.e.* Self_Guid) should also boost generalization because it enforces the target model to learn more information and focus more on the hard sample from itself;

Results are illustrated in Tab. 7. We observe that both the two variants LFME_Guid and Self_Guid are beneficial to the baseline models, leading ERM in all unseen domains. These results further validate our first analysis in Sec. 4, and we can conclude that when the basic discrimination is not compromised, either directly or indirectly enlarging $q_c$ for $\forall c \neq *$ can help the target model learn more information, and improve generalization accordingly.

### D.3 Hard Samples from the Experts

As analyzed in Sec. 4, LFME not only encourages the target model to learn more information but also implicitly helps it to focus more on the hard samples from the experts. To examine if the classifications of hard samples from the experts are indeed improved, we plot the classification ratio $\mathcal{R} = \frac{\overline{p}_*}{\max(\overline{p})}$, where $\overline{p}$ denotes the normalized probability: $\overline{p} = \frac{p - \min(p)}{\max(p) - \min(p)}$, from the target model on these hard samples. We use the ratio because it can quantify the correct predictions regarding the hard samples more precisely than the accuracy: the closer $\mathcal{R}$ approaches 1, the better the corresponding model performs on the sample. In this setting, $1/3$ samples with larger losses in a training batch are considered hard samples. We compare $\mathcal{R}$ from LFME with that from ERM to highlight the difference. As can be observed in Fig. 5 (a), most $\mathcal{R}$ from LFME is larger than that from ERM over the iterations, denoting LFME is better at handling these hard samples than ERM. As a comparison, $\mathcal{R}$ from these two models almost overlap in the easy samples as shown in Fig. 5 (b). These results validate our analysis that LFME implicitly helps the target model to focus more on the hard samples compared with ERM.

Meanwhile, to validate if the hard samples from experts can indeed help generalization, we conduct experiments by explicitly putting more weights on the hard samples from the experts on the basis of the free launch ERM+ (*i.e.* ERM+w/ expt). Specifically, the target objective based on Eq. (9) is adjusted into: $w\mathcal{L}_{all}$, where $w$ is the weight for the training samples and is determined based on the loss of the experts. Basically, the larger the corresponding loss from the experts, the larger the value of $w$ should be, and we use the strategy from [35] to implement this hard sample mining process. As can be observed in Tab. 7, mining hard samples from the experts can improve the base model of ERM+, especially in the difficult TerraInc dataset. These results validate that mining hard samples from experts is beneficial for generalization.

### D.4 Comparisons Between Hard Samples from Different Models: Why Experts are Required

We note in Tab. 7, the model Self_Guid performs inferior to LFME in both the two datasets, and the improvements are also marginal compared with ERM+ on account of variances. These results indicate the hard samples from the model itself may be less effective for generalization compared with that from the experts. To validate this hypothesis, we conduct experiments by explicitly focusing more on the hard samples from the model itself on the basis of ERM+ (*i.e.* ERM+ w/ self). The implementation is the same as that in ERM+ w/ expt.

Results listed in the 6th row in Tab. 7 validate this hypothesis, where the hard samples from the model itself can hardly improve generalization compared with that from the experts. We presume the main reason is that besides the dominant specific domain information, hard samples from the experts also contain some out-of-the-domain information that makes it challenging for the corresponding expert. Examples are shown in Fig. 6. We observe that compared with hard samples from the model itself (*i.e.* Fig. 6 (b)), hard samples from the experts (*i.e.* Fig. 6 (a)) contain more ambiguous data located in the mixed region or the boundary of two domains than, indicating the experts may be more effective at revealing samples that with characteristics from different domains. By implicitly emphasizing these out-of-the-specific-domain samples, can the target model learn domain-agnostic features, thus improving generalization accordingly. Because the naive training strategies are fed with data from different domains at the same time, the out-of-the-specific-domain information is difficult to discover in their framework, explaining why hard samples from other strategies may not contribute. This is also the reason why the experts must be involved in the overall framework.

## E Compare with MoE-Based Ideas and Other Arts

Since our idea involves training multiple experts, one may connect it to the idea of mixture of experts (MoE) [86]. However, it is noteworthy that our LFME is not a simple implementation of existing techniques. The introduced logit regularization term is new and can be properly explained for its effectiveness. Some similar ideas that use MoE have been explored in other works for improving DG. For example, in DAELDG [90], following a shared feature extractor, each domain corresponds to a specific classifier (*i.e.* expert), whose output is enforced to be similar to the average outputs from the non-expert classifiers. Different from our work, it uses the average outputs from different experts as the final result. In Meta-DMoE [89], a meta-learning-based framework is employed to enforce

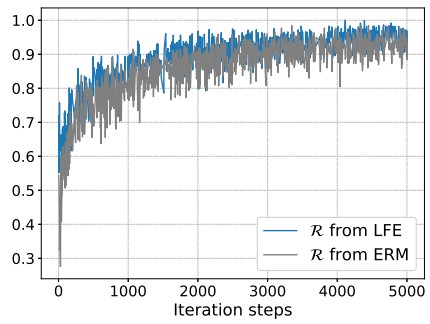
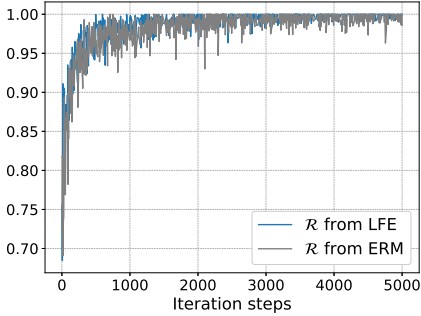

(a) $\mathcal{R}$ from the hard samples

(b) $\mathcal{R}$ from the easy samples

Figure 5: Classification ratio comparisons of ERM and LFME in the hard and easy samples from the difficult TerraInc dataset. The closer the ratio $\mathcal{R}$ approaches 1, the better the corresponding prediction. Here the hard samples are specified by the experts: the $1/3$ samples in a training batch with larger losses from the experts, and the easy samples are the leading $1/3$ samples with smaller losses. The two models perform evenly well on the easy samples, while LFME obtains better results than ERM in the hard samples.

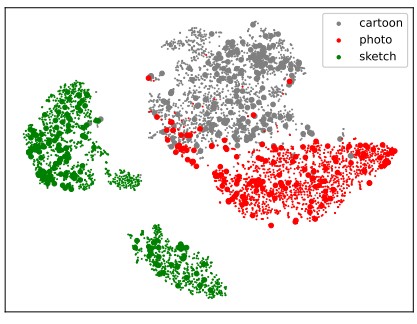
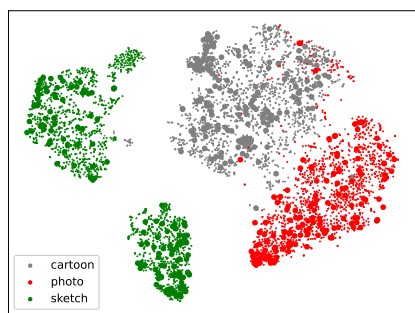

(a) Hard samples from experts

(b) Hard samples from the model itself

Figure 6: T-SNE visualizations of hard samples (larger dots) from different models. Here the data from different domains are clustered by their styles [91] (*i.e.* feature statistics from the first layer in the same ResNet18 model), and 10 percents of samples with larger loss are considered hard samples in a domain. Hard samples from the experts contain more ambiguous data located in the mixed region or the boundary of two different domains.

the feature of the target model to be similar to the aggregated features from the experts, which is different from our logit regularization idea. In Sec. D.1 and 6.3, we implement the basic designs of these two ideas with the framework of LFME and show their ineffectiveness. Besides the comparison with Meta-DMoE in Tab. 1, in this section, we further compare LFME with them using their original settings in the PACS and OfficeHome datasets with the benchmark provided in [90]. We also compare with several other recent DG methods. Results in Tab. 8 show that LFME performs favorably against DAELDG, and outperforms Meta-DMoE, further validating the advantages of the logit regularization term. We also note LFME performs better than other arts with a different benchmark, demonstrating the effectiveness of the proposed approach.

## F Existing Ideas with Same Training Resources

Since LFME requires learning different experts during training, inevitably using more parameters than existing methods, one may wonder if the effectiveness of LFME derives from using these extra parameters from the experts. To examine this idea, we conduct experiments by implementing some leading arts (*i.e.* CORAL, SD) and the baseline ERM with $M + 1$ times of model size (where $M$ is the domain number). Specifically, the feature extractor for the larger method will contain $M + 1$ branches, each with the same pretrained ResNet backbone. We concate the final outputs from the different branches and use it as input for a classifier to obtain the final result. Note that in this setting, a sample

Table 8: Out-of-domain evaluations of other related methods with the benchmark provided in [90].

| Model | Target domain in PACS | | | | Avg. | Target domain in OfficeHome | | | | Avg. |
|---|---|---|---|---|---|---|---|---|---|---|
| | Art | Cartoon | Photo | Sketch | | Art | Clipart | Product | Real | |
| Compared with other recent methods | | | | | | | | | | |
| ERM | 77.0 | 75.9 | 96.0 | 69.2 | 79.5 | 58.9 | 49.4 | 74.3 | 76.2 | 64.7 |
| CCSA [52] | 80.5 | 76.9 | 93.6 | 66.8 | 79.4 | 59.9 | 49.9 | 74.1 | 75.7 | 64.9 |
| JiGen [7] | 79.4 | 75.3 | 96.0 | 71.6 | 80.5 | 53.0 | 47.5 | 71.5 | 72.8 | 61.2 |
| CrossGrad [67] | 79.8 | 76.8 | 96.0 | 70.2 | 80.7 | 58.4 | 49.4 | 73.9 | 75.8 | 64.4 |
| Epi-FCR [45] | 82.1 | 77.0 | 93.9 | 73.0 | 81.5 | - | - | - | - | - |
| DMG [10] | 76.9 | 80.4 | 93.4 | 75.2 | 81.5 | - | - | - | - | - |
| Compared with MoE-based methods | | | | | | | | | | |
| DAELDG [90] | 84.6 | 74.4 | 95.6 | 78.9 | 83.4 | 59.4 | 55.1 | 74.0 | 75.7 | 66.1 |
| Meta-DMoE [89] | 83.2 | 76.8 | 95.4 | 76.6 | 83.0 | 58.9 | 55.5 | 73.6 | 74.4 | 65.6 |
| Ours | 83.4 | 78.3 | 96.8 | 76.1 | 83.7 | 60.4 | 55.7 | 74.6 | 75.1 | 66.5 |

Table 9: Existing methods with same resources during training. We expand existing ideas by concatenating $M+1$ branches to ensure they use the same parameters as LFME during training.

| Model | Target domain | | | | Avg. |
|---|---|---|---|---|---|
| | Art | Cartoon | Photo | Sketch | |
| Performances from original designs | | | | | |
| ERM | 78.0 ± 1.3 | 73.4 ± 0.8 | 94.1 ± 0.4 | 73.6 ± 2.2 | 79.8 ± 0.4 |
| CORAL | 81.5 ± 0.5 | 75.4 ± 0.7 | 95.2 ± 0.5 | 74.8 ± 0.4 | 81.7 ± 0.0 |
| SD | 83.2 ± 0.6 | 74.6 ± 0.3 | 94.6 ± 0.1 | 75.1 ± 1.6 | 81.9 ± 0.3 |
| Performances from $M+1$ model sizes | | | | | |
| ERM | 76.6 ± 1.1 | 75.1 ± 1.3 | 94.5 ± 0.2 | 73.1 ± 1.7 | 79.8 ± 0.4 |
| CORAL | 81.2 ± 0.5 | 75.8± 0.4 | 95.4 ± 0.2 | 75.4±0.7 | 81.9 ± 0.1 |
| SD | 81.6 ± 0.8 | 75.4 ± 0.5 | 94.7 ± 0.1 | 76.9 ± 0.8 | 82.2 ± 0.4 |
| Ours | 81.0 ± 0.9 | 76.5 ± 0.9 | 94.6 ± 0.5 | 77.4 ± 0.2 | 82.4 ± 0.1 |

will go through $M+1$ times of forward passes for both training and inference, which is more than that of LFME design. Results are shown in Tab. 9. We note that compared to the results from the original models, when using the same pretrained knowledge, naively expanding model size cannot improve the performance. The reason may be that a well-pretrained small backbone can already saturate on limited training data (as shown in Table 5, ERM can achieve more than 0.96 acc in the source domains), thus it is unnecessary to use more parameters in these datasets.

## G   Detailed Settings of Dyn_Expt

Besides the different experts $E_i$ where $i$ denotes the domain label, Dyn_Expt also uses an extra weighting network $W$ to estimate the labels of the training data. During the training stage, the experts are trained the same as that in LFME, and the weighting network is trained using the corresponding domain labels by minimizing $\mathcal{H}(W(x^i), i)$. During the test phase, Dyn_Expt dynamically combines the outputs from the experts with the corresponding domain probability: for $\forall x \in \mathcal{D}_{M+1}$, the final result is $\sum_i^M W(x)_i E_i(x)$. We use the same backbone for both the experts and the weighting module in this experiment.

## H   Detailed Results

This section presents the detailed results on the semantic segmentation task in Tab. 10, and detailed results in the unseen domains from the different unseen domains of DomainBed benchmark in Tab. 11, 12, 13 14, and 15.

Table 10: Evaluations on the semantic segmentation task with the metrics of Mean IoU (%) and per-class IoU (%). Source data is from the synthetic GTAV [63] and Synthia [64] datasets, and the target data is from the real-world Cityscapes [20], BDD100K [83], and the Mapillary [55] datasets. All models are with the same backbone: DeepLabV3+ with ResNet50, and results with † are from [19].

| | Methods | road | sidewalk | building | wall | fence | pole | t-light | t-sign | vegetation | terrain | sky | person | rider | car | truck | bus | train | m-bike | bicycle | mIoU |
|---|---|---|---|---|---|---|---|---|---|---|---|---|---|---|---|---|---|---|---|---|---|
| Cityscapes | Baseline† | 72.7 | 36.4 | 64.9 | 11.9 | 2.8 | 31.0 | 37.7 | 20.0 | 84.9 | 14.0 | 71.9 | 65.3 | 9.9 | 84.7 | 11.6 | 25.4 | 0.0 | 10.6 | 18.1 | 35.46 |
| | IBN-Net† | 68.3 | 29.5 | 69.7 | 17.4 | 1.8 | 30.7 | 36.2 | 20.2 | 85.4 | 18.2 | 81.8 | 64.7 | 12.9 | 82.7 | 13.0 | 16.2 | 0.0 | 8.2 | 22.2 | 35.55 |
| | RobustNet† | 82.6 | 40.1 | 73.4 | 17.4 | 1.4 | 34.2 | 38.6 | 18.5 | 84.9 | 16.9 | 81.9 | 65.2 | 11.4 | 84.7 | 7.2 | 23.6 | 0.0 | 10.4 | 23.9 | 37.69 |
| | Baseline | 65.4 | 34.1 | 64.0 | 20.8 | 20.7 | 28.8 | 41.3 | 23.4 | 83.5 | 33.7 | 58.2 | 64.9 | 13.9 | 69.6 | 23.5 | 14.7 | 9.8 | 17.3 | 19.4 | 37.19 |
| | PinMem | 84.6 | 43.3 | 79.0 | 20.3 | 7.5 | 38.1 | 39.3 | 23.4 | 86.0 | 24.6 | 69.8 | 66.5 | 18.9 | 82.1 | 25.6 | 35.7 | 3.1 | 25.8 | 21.5 | 41.86 |
| | SD | 52.9 | 33.6 | 55.2 | 21.3 | 19.2 | 29.1 | 41.2 | 22.6 | 83.7 | 35.0 | 54.9 | 64.8 | 12.3 | 61.6 | 23.6 | 14.3 | 9.4 | 14.4 | 11.7 | 34.77 |
| | Ours | 71.9 | 32.7 | 69.9 | 19.4 | 20.3 | 30.5 | 34.5 | 16.7 | 84.0 | 30.0 | 82.8 | 67.0 | 21.7 | 67.1 | 30.7 | 15.4 | 1.9 | 16.0 | 16.7 | 38.38 |
| BDD100K | Baseline† | 44.6 | 26.1 | 34.7 | 1.8 | 6.9 | 29.5 | 39.1 | 20.5 | 64.9 | 10.8 | 51.6 | 50.6 | 10.2 | 63.9 | 1.1 | 4.8 | 0.0 | 5.5 | 10.1 | 25.09 |
| | IBN-Net† | 53.8 | 25.0 | 55.4 | 2.8 | 14.8 | 32.9 | 39.7 | 26.3 | 71.7 | 16.4 | 85.9 | 57.4 | 17.5 | 56.9 | 5.3 | 6.0 | 0.0 | 18.5 | 25.4 | 32.18 |
| | RobustNet† | 69.5 | 35.0 | 60.9 | 4.1 | 13.1 | 36.6 | 40.5 | 27.3 | 71.6 | 14.0 | 83.6 | 56.0 | 17.3 | 61.9 | 4.4 | 8.8 | 0.0 | 24.3 | 18.9 | 34.09 |
| | Baseline | 57.1 | 27.3 | 37.3 | 2.8 | 20.5 | 29.8 | 36.8 | 22.1 | 59.0 | 23.2 | 35.1 | 48.4 | 9.5 | 70.6 | 15.3 | 8.5 | 0.0 | 19.3 | 8.5 | 27.95 |
| | PinMem | 73.4 | 36.8 | 56.7 | 4.7 | 25.1 | 32.8 | 37.5 | 24.1 | 71.7 | 23.2 | 72.1 | 53.6 | 17.4 | 68.1 | 9.4 | 29.1 | 0.0 | 16.7 | 11.6 | 34.94 |
| | SD | 56.6 | 29.6 | 39.6 | 2.8 | 17.4 | 31.8 | 37.1 | 19.6 | 60.3 | 23.4 | 36.6 | 49.6 | 10.5 | 67.1 | 15.1 | 9.3 | 0.0 | 19.9 | 5.5 | 28.00 |
| | Ours | 72.2 | 30.5 | 58.4 | 8.5 | 30.1 | 30.5 | 32.6 | 21.5 | 69.0 | 23.7 | 75.7 | 54.7 | 18.5 | 68.1 | 18.6 | 26.9 | 0.0 | 29.2 | 9.7 | 35.70 |
| Mapillary | Baseline† | 62.0 | 36.3 | 32.5 | 9.5 | 7.7 | 29.9 | 40.5 | 22.5 | 78.6 | 40.9 | 61.0 | 59.4 | 6.4 | 78.3 | 5.1 | 5.1 | 0.1 | 9.0 | 21.8 | 31.94 |
| | IBN-Net† | 67.4 | 38.8 | 51.3 | 10.2 | 7.6 | 36.0 | 40.1 | 40.8 | 80.3 | 39.9 | 92.1 | 61.8 | 14.0 | 74.4 | 10.7 | 9.4 | 3.5 | 15.3 | 25.4 | 38.09 |
| | RobustNet† | 78.0 | 41.0 | 56.6 | 13.1 | 6.2 | 39.4 | 41.3 | 36.1 | 79.5 | 34.7 | 90.0 | 61.0 | 12.0 | 76.1 | 10.7 | 13.1 | 0.8 | 16.9 | 24.8 | 38.49 |
| | Baseline | 58.1 | 31.4 | 34.7 | 9.0 | 18.0 | 30.8 | 38.3 | 15.1 | 69.8 | 30.3 | 54.3 | 55.7 | 8.6 | 77.7 | 22.9 | 6.0 | 7.6 | 18.3 | 21.8 | 32.01 |
| | PinMem | 76.0 | 40.8 | 48.7 | 14.5 | 15.3 | 36.6 | 38.4 | 41.8 | 79.8 | 33.1 | 77.3 | 62.0 | 17.6 | 74.2 | 26.5 | 19.0 | 6.0 | 18.7 | 22.6 | 39.41 |
| | SD | 58.3 | 31.8 | 37.8 | 9.0 | 12.6 | 31.5 | 38.1 | 9.8 | 68.7 | 30.6 | 56.3 | 55.5 | 8.4 | 72.7 | 23.6 | 7.5 | 7.5 | 17.3 | 18.0 | 31.41 |
| | Ours | 72.3 | 36.4 | 62.0 | 15.1 | 26.2 | 38.5 | 39.1 | 45.7 | 73.4 | 30.4 | 91.0 | 59.3 | 15.8 | 74.0 | 34.2 | 18.6 | 7.7 | 21.6 | 18.2 | 41.04 |

Table 11: Average accuracies on the PACS [43] datasets using the default hyper-parameter settings in DomainBed [27]. TT denotes the average training time (minutes) for one trial in a target domain.

| | art | cartoon | photo | sketch | Average | TT |
|---|---|---|---|---|---|---|
| ERM [75] | 78.0 ± 1.3 | 73.4 ± 0.8 | 94.1 ± 0.4 | 73.6 ± 2.2 | 79.8 ± 0.4 | 24 |
| IRM [1] | 76.9 ± 2.6 | 75.1 ± 0.7 | 94.3 ± 0.4 | 77.4 ± 0.4 | 80.9 ± 0.5 | 18 |
| GroupGRO [66] | 77.7 ± 2.6 | 76.4 ± 0.3 | 94.0 ± 0.3 | 74.8 ± 1.3 | 80.7 ± 0.4 | 24 |
| Mixup [80] | 79.3 ± 1.1 | 74.2 ± 0.3 | 94.9 ± 0.3 | 68.3 ± 2.7 | 79.2 ± 0.9 | 18 |
| MLDG [44] | 78.4 ± 0.7 | 75.1 ± 0.5 | 94.8 ± 0.4 | 76.7 ± 0.8 | 81.3 ± 0.2 | 32 |
| CORAL [70] | 81.5 ± 0.5 | 75.4 ± 0.7 | 95.2 ± 0.5 | 74.8 ± 0.4 | 81.7 ± 0.0 | 24 |
| MMD [46] | 81.3 ± 0.6 | 75.5 ± 1.0 | 94.0 ± 0.5 | 74.3 ± 1.5 | 81.3 ± 0.8 | 18 |
| DANN [24] | 79.0 ± 0.6 | 72.5 ± 0.7 | 94.4 ± 0.5 | 70.8 ± 3.0 | 79.2 ± 0.3 | 17 |
| CDANN [49] | 80.4 ± 0.8 | 73.7 ± 0.3 | 93.1 ± 0.6 | 74.2 ± 1.7 | 80.3 ± 0.5 | 24 |
| MTL [5] | 78.7 ± 0.6 | 73.4 ± 1.0 | 94.1 ± 0.6 | 74.4 ± 3.0 | 80.1 ± 0.8 | 18 |
| SagNet [54] | 82.9 ± 0.4 | 73.2 ± 1.1 | 94.6 ± 0.5 | 76.1 ± 1.8 | 81.7 ± 0.6 | 24 |
| ARM [87] | 79.4 ± 0.6 | 75.0 ± 0.7 | 94.3 ± 0.6 | 73.8 ± 0.6 | 80.6 ± 0.5 | 24 |
| VREx [42] | 74.4 ± 0.7 | 75.0 ± 0.4 | 93.3 ± 0.3 | 78.1 ± 0.9 | 80.2 ± 0.5 | 17 |
| RSC [36] | 78.5 ± 1.1 | 73.3 ± 0.9 | 93.6 ± 0.6 | 76.5 ± 1.4 | 80.5 ± 0.2 | 25 |
| Meta-DMoE [89] | 78.7 ± 0.5 | 74.2 ± 1.1 | 94.4 ± 0.3 | 76.7 ± 0.9 | 81.0 ± 0.3 | 45 |
| SelfReg [39] | 82.5 ± 0.8 | 74.4 ± 1.5 | 95.4 ± 0.5 | 74.9 ± 1.3 | 81.8 ± 0.3 | 25 |
| MIRO [9] | 79.3 ± 0.6 | 68.1 ± 2.5 | 95.5 ± 0.3 | 60.6 ± 3.1 | 75.9 ± 1.4 | 31 |
| MixStyle [91] | 82.6 ± 1.2 | 76.3 ± 0.4 | 94.2 ± 0.3 | 77.5 ± 1.3 | 82.6 ± 0.4 | 25 |
| Fish [69] | 80.9 ± 1.0 | 75.9 ± 0.4 | 95.0 ± 0.4 | 76.2 ± 1.0 | 82.0 ± 0.3 | 52 |
| SD [60] | 83.2 ± 0.6 | 74.6 ± 0.3 | 94.6 ± 0.1 | 75.1 ± 1.6 | 81.9 ± 0.3 | 25 |
| CAD [65] | 83.9 ± 0.8 | 74.2 ± 0.4 | 94.6 ± 0.4 | 75.0 ± 1.2 | 81.9 ± 0.3 | 24 |
| CondCAD [65] | 79.7 ± 1.0 | 74.2 ± 0.9 | 94.6 ± 0.4 | 74.8 ± 1.4 | 80.8 ± 0.5 | 26 |
| Fishr [62] | 81.2 ± 0.4 | 75.8 ± 0.8 | 94.3 ± 0.3 | 73.8 ± 0.6 | 81.3 ± 0.3 | 17 |
| ITTA [15] | 84.7 ± 0.4 | 78.0 ± 0.4 | 94.5 ± 0.4 | 78.2 ± 0.3 | 83.8 ± 0.3 | 62 |
| ERM+ | 81.9 ± 0.4 | 75.1 ± 0.7 | 94.8 ± 0.7 | 73.8 ± 2.2 | 81.4 ± 0.5 | 24 |
| Ours | 81.0 ± 0.9 | 76.5 ± 0.9 | 94.6 ± 0.5 | 77.4 ± 0.2 | 82.4 ± 0.1 | 38 |

Table 12: Average accuracies on the VLCS [23] datasets using the default hyper-parameter settings in DomainBed [27].

|  | Caltech | LabelMe | Sun | VOC | Average |
|---|---|---|---|---|---|
| ERM [75] | 97.7 ± 0.3 | 62.1 ± 0.9 | 70.3 ± 0.9 | 73.2 ± 0.7 | 75.8 ± 0.2 |
| IRM [1] | 96.1 ± 0.8 | 62.5 ± 0.3 | 69.9 ± 0.7 | 72.0 ± 1.4 | 75.1 ± 0.1 |
| GroupGRO [66] | 96.7 ± 0.6 | 61.7 ± 1.5 | 70.2 ± 1.8 | 72.9 ± 0.6 | 75.4 ± 1.0 |
| Mixup [80] | 95.6 ± 1.5 | 62.7 ± 0.4 | 71.3 ± 0.3 | 75.4 ± 0.2 | 76.2 ± 0.3 |
| MLDG [44] | 95.8 ± 0.5 | 63.3 ± 0.8 | 68.5 ± 0.5 | 73.1 ± 0.8 | 75.2 ± 0.3 |
| CORAL [70] | 96.5 ± 0.3 | 62.8 ± 0.1 | 69.1 ± 0.6 | 73.8 ± 1.0 | 75.5 ± 0.4 |
| MMD [46] | 96.0 ± 0.8 | 64.3 ± 0.6 | 68.5 ± 0.6 | 70.8 ± 0.1 | 74.9 ± 0.5 |
| DANN [24] | 97.2 ± 0.1 | 63.3 ± 0.6 | 70.2 ± 0.9 | 74.4 ± 0.2 | 76.3 ± 0.2 |
| CDANN [49] | 95.4 ± 1.2 | 62.6 ± 0.6 | 69.9 ± 1.3 | 76.2 ± 0.5 | 76.0 ± 0.5 |
| MTL [5] | 94.4 ± 2.3 | 65.0 ± 0.6 | 69.6 ± 0.6 | 71.7 ± 1.3 | 75.2 ± 0.3 |
| SagNet [54] | 94.9 ± 0.7 | 61.9 ± 0.7 | 69.6 ± 1.3 | 75.2 ± 0.6 | 75.4 ± 0.8 |
| ARM [87] | 96.9 ± 0.5 | 61.9 ± 0.4 | 71.6 ± 0.1 | 73.3 ± 0.4 | 75.9 ± 0.3 |
| VREx [42] | 96.2 ± 0.0 | 62.5 ± 1.3 | 69.3 ± 0.9 | 73.1 ± 1.2 | 75.3 ± 0.6 |
| RSC [36] | 96.2 ± 0.0 | 63.6 ± 1.3 | 69.8 ± 1.0 | 72.0 ± 0.4 | 75.4 ± 0.3 |
| Meta-DMoE [89] | 96.4 ± 0.2 | 62.5 ± 1.0 | 70.3 ± 0.3 | 74.9 ± 1.1 | 76.0 ± 0.6 |
| SelfReg [39] | 95.8 ± 0.6 | 63.4 ± 1.1 | 71.1 ± 0.6 | 75.3 ± 0.6 | 76.4 ± 0.7 |
| MixStyle [91] | 97.3 ± 0.3 | 61.6 ± 0.1 | 70.4 ± 0.7 | 71.3 ± 1.9 | 75.2 ± 0.7 |
| Fish [69] | 97.4 ± 0.2 | 63.4 ± 0.1 | 71.5 ± 0.4 | 75.2 ± 0.7 | 76.9 ± 0.2 |
| SD [60] | 96.5 ± 0.4 | 62.2 ± 0.0 | 69.7 ± 0.9 | 73.6 ± 0.4 | 75.5 ± 0.4 |
| CAD [65] | 94.5 ± 0.9 | 63.5 ± 0.6 | 70.4 ± 1.2 | 72.4 ± 1.3 | 75.2 ± 0.6 |
| CondCAD [65] | 96.5 ± 0.8 | 62.6 ± 0.4 | 69.1 ± 0.2 | 76.0 ± 0.2 | 76.1 ± 0.3 |
| Fishr [62] | 97.2 ± 0.6 | 63.3 ± 0.7 | 70.4 ± 0.6 | 74.0 ± 0.8 | 76.2 ± 0.3 |
| ITTA [15] | 96.9 ± 1.2 | 63.7 ± 1.1 | 72.0 ± 0.3 | 74.9 ± 0.8 | 76.9 ± 0.6 |
| ERM+ | 96.0 ± 0.3 | 61.9 ± 0.4 | 71.5 ± 0.4 | 75.0 ± 1.2 | 76.1 ± 0.4 |
| Ours | 96.4 ± 0.3 | 62.8 ± 1.1 | 70.1 ± 0.3 | 75.4 ± 0.8 | 76.2 ± 0.4 |

Table 13: Average accuracies on the OfficeHome [76] datasets using the default hyper-parameter settings in DomainBed [27].

|  | art | clipart | product | real | Average |
|---|---|---|---|---|---|
| ERM [75] | 52.2 ± 0.2 | 48.7 ± 0.5 | 69.9 ± 0.5 | 71.7 ± 0.5 | 60.6 ± 0.2 |
| IRM [1] | 49.7 ± 0.2 | 46.8 ± 0.5 | 67.5 ± 0.4 | 68.1 ± 0.6 | 58.0 ± 0.1 |
| GroupGRO [66] | 52.6 ± 1.1 | 48.2 ± 0.9 | 69.9 ± 0.4 | 71.5 ± 0.8 | 60.6 ± 0.3 |
| Mixup [80] | 54.0 ± 0.7 | 49.3 ± 0.7 | 70.7 ± 0.7 | 72.6 ± 0.3 | 61.7 ± 0.5 |
| MLDG [44] | 53.1 ± 0.3 | 48.4 ± 0.3 | 70.5 ± 0.7 | 71.7 ± 0.4 | 60.9 ± 0.2 |
| CORAL [70] | 55.1 ± 0.7 | 49.7 ± 0.9 | 71.8 ± 0.2 | 73.1 ± 0.5 | 62.4 ± 0.4 |
| MMD [46] | 50.9 ± 1.0 | 48.7 ± 0.3 | 69.3 ± 0.7 | 70.7 ± 1.3 | 59.9 ± 0.4 |
| DANN [24] | 51.8 ± 0.5 | 47.1 ± 0.1 | 69.1 ± 0.7 | 70.2 ± 0.7 | 59.5 ± 0.5 |
| CDANN [49] | 51.4 ± 0.5 | 46.9 ± 0.6 | 68.4 ± 0.5 | 70.4 ± 0.4 | 59.3 ± 0.4 |
| MTL [5] | 51.6 ± 1.5 | 47.7 ± 0.5 | 69.1 ± 0.3 | 71.0 ± 0.6 | 59.9 ± 0.5 |
| SagNet [54] | 55.3 ± 0.4 | 49.6 ± 0.2 | 72.1 ± 0.4 | 73.2 ± 0.4 | 62.5 ± 0.3 |
| ARM [87] | 51.3 ± 0.9 | 48.5 ± 0.4 | 68.0 ± 0.3 | 70.6 ± 0.1 | 59.6 ± 0.3 |
| VREx [42] | 51.1 ± 0.3 | 47.4 ± 0.6 | 69.0 ± 0.4 | 70.5 ± 0.4 | 59.5 ± 0.1 |
| RSC [36] | 49.0 ± 0.1 | 46.2 ± 1.5 | 67.8 ± 0.7 | 70.6 ± 0.3 | 58.4 ± 0.6 |
| Meta-DMoE [89] | 54.7 ± 0.3 | 50.4 ± 0.9 | 71.8 ± 0.3 | 71.8 ± 0.1 | 62.2 ± 0.1 |
| SelfReg [39] | 55.1 ± 0.8 | 49.2 ± 0.6 | 72.2 ± 0.3 | 73.0 ± 0.3 | 62.4 ± 0.1 |
| MixStyle [91] | 50.8 ± 0.6 | 51.4 ± 1.1 | 67.6 ± 1.3 | 68.8 ± 0.5 | 59.6 ± 0.8 |
| Fish [69] | 54.6 ± 1.0 | 49.6 ± 1.0 | 71.3 ± 0.6 | 72.4 ± 0.2 | 62.0 ± 0.6 |
| SD [60] | 55.0 ± 0.4 | 51.3 ± 0.5 | 72.5 ± 0.2 | 72.7 ± 0.3 | 62.9 ± 0.2 |
| CAD [65] | 52.1 ± 0.6 | 48.3 ± 0.5 | 69.7 ± 0.3 | 71.9 ± 0.4 | 60.5 ± 0.3 |
| CondCAD [65] | 53.3 ± 0.6 | 48.4 ± 0.2 | 69.8 ± 0.9 | 72.6 ± 0.1 | 61.0 ± 0.4 |
| Fishr [62] | 52.6 ± 0.9 | 48.6 ± 0.3 | 69.9 ± 0.6 | 72.4 ± 0.4 | 60.9 ± 0.3 |
| ITTA [15] | 54.4 ± 0.2 | 52.3 ± 0.8 | 69.5 ± 0.3 | 71.7 ± 0.2 | 62.0 ± 0.2 |
| ERM+ | 56.1 ± 0.3 | 51.0 ± 0.3 | 73.0 ± 0.3 | 72.5 ± 0.2 | 63.2 ± 0.1 |
| Ours | 56.4 ± 0.1 | 51.1 ± 0.5 | 72.5 ± 0.2 | 72.8 ± 0.1 | 63.2 ± 0.1 |

Table 14: Average accuracies on the TerraInc [3] datasets using the default hyper-parameter settings in DomainBed [27].

| | L100 | L38 | L43 | L46 | Average |
|---|---|---|---|---|---|
| ERM [75] | 42.1 ± 2.5 | 30.1 ± 1.2 | 48.9 ± 0.6 | 34.0 ± 1.1 | 38.8 ± 1.0 |
| IRM [1] | 41.8 ± 1.8 | 29.0 ± 3.6 | 49.6 ± 2.1 | 33.1 ± 1.5 | 38.4 ± 0.9 |
| GroupGRO [66] | 45.3 ± 4.6 | 36.1 ± 4.4 | 51.0 ± 0.8 | 33.7 ± 0.9 | 41.5 ± 2.0 |
| Mixup [80] | 49.4 ± 2.0 | 35.9 ± 1.8 | 53.0 ± 0.7 | 30.0 ± 0.9 | 42.1 ± 0.7 |
| MLDG [44] | 39.6 ± 2.3 | 33.2 ± 2.7 | 52.4 ± 0.5 | 35.1 ± 1.5 | 40.1 ± 0.9 |
| CORAL [70] | 46.7 ± 3.2 | 36.9 ± 4.3 | 49.5 ± 1.9 | 32.5 ± 0.7 | 41.4 ± 1.8 |
| MMD [46] | 49.1 ± 1.2 | 36.4 ± 4.8 | 50.4 ± 2.1 | 32.3 ± 1.5 | 42.0 ± 1.0 |
| DANN [24] | 44.3 ± 3.6 | 28.0 ± 1.5 | 47.9 ± 1.0 | 31.3 ± 0.6 | 37.9 ± 0.9 |
| CDANN [49] | 36.9 ± 6.4 | 32.7 ± 6.2 | 51.1 ± 1.3 | 33.5 ± 0.5 | 38.6 ± 2.3 |
| MTL [5] | 45.2 ± 2.6 | 31.0 ± 1.6 | 50.6 ± 1.1 | 34.9 ± 0.4 | 40.4 ± 1.0 |
| SagNet [54] | 36.3 ± 4.7 | 40.3 ± 2.0 | 52.5 ± 0.6 | 33.3 ± 1.3 | 40.6 ± 1.5 |
| ARM [87] | 41.5 ± 4.5 | 27.7 ± 2.4 | 50.9 ± 1.0 | 29.6 ± 1.5 | 37.4 ± 1.9 |
| VREx [42] | 48.0 ± 1.7 | 41.1 ± 1.5 | 51.8 ± 1.5 | 32.0 ± 1.2 | 43.2 ± 0.3 |
| RSC [36] | 42.8 ± 2.4 | 32.2 ± 3.8 | 49.6 ± 0.9 | 32.9 ± 1.2 | 39.4 ± 1.3 |
| Meta-DMoE [89] | 47.1 ± 2.0 | 29.2 ± 2.1 | 47.8 ± 0.8 | 35.8 ± 1.8 | 40.0 ± 1.2 |
| SelfReg [39] | 46.1 ± 1.5 | 34.5 ± 1.6 | 49.8 ± 0.3 | 34.7 ± 1.5 | 41.3 ± 0.3 |
| MixStyle [91] | 50.6 ± 1.9 | 28.0 ± 4.5 | 52.1 ± 0.7 | 33.0 ± 0.2 | 40.9 ± 1.1 |
| Fish [69] | 46.3 ± 3.0 | 29.0 ± 1.1 | 52.7 ± 1.2 | 32.8 ± 1.0 | 40.2 ± 0.6 |
| SD [60] | 45.5 ± 1.9 | 33.2 ± 3.1 | 52.9 ± 0.7 | 36.4 ± 0.8 | 42.0 ± 1.0 |
| CAD [65] | 43.1 ± 2.6 | 31.1 ± 1.9 | 53.1 ± 1.6 | 34.7 ± 1.3 | 40.5 ± 0.4 |
| CondCAD [65] | 44.4 ± 2.9 | 32.9 ± 2.5 | 50.5 ± 1.3 | 30.8 ± 0.5 | 39.7 ± 0.4 |
| Fishr [62] | 49.9 ± 3.3 | 36.6 ± 0.9 | 49.8 ± 0.2 | 34.2 ± 1.3 | 42.6 ± 1.0 |
| ITTA [15] | 51.7 ± 2.4 | 37.6 ± 0.6 | 49.9 ± 0.6 | 33.6 ± 0.6 | 43.2 ± 0.5 |
| ERM+ | 46.7 ± 2.6 | 37.1 ± 1.3 | 53.2 ± 0.4 | 34.8 ± 1.3 | 42.9 ± 0.7 |
| Ours | 53.4 ± 0.4 | 40.7 ± 2.4 | 54.9 ± 0.4 | 36.4 ± 0.7 | 46.3 ± 0.5 |

Table 15: Average accuracies on the DomainNet [59] datasets using the default hyper-parameter settings in DomainBed [27].

| | clip | info | paint | quick | real | sketch | Average |
|---|---|---|---|---|---|---|---|
| ERM [75] | 50.4 ± 0.2 | 14.0 ± 0.2 | 40.3 ± 0.5 | 11.7 ± 0.2 | 52.0 ± 0.2 | 43.2 ± 0.3 | 35.3 ± 0.1 |
| IRM [1] | 43.2 ± 0.9 | 12.6 ± 0.3 | 35.0 ± 1.4 | 9.9 ± 0.4 | 43.4 ± 3.0 | 38.4 ± 0.4 | 30.4 ± 1.0 |
| GroupGRO [66] | 38.2 ± 0.5 | 13.0 ± 0.3 | 28.7 ± 0.3 | 8.2 ± 0.1 | 43.4 ± 0.5 | 33.7 ± 0.0 | 27.5 ± 0.1 |
| Mixup [80] | 48.9 ± 0.3 | 13.6 ± 0.3 | 39.5 ± 0.5 | 10.9 ± 0.4 | 49.9 ± 0.2 | 41.2 ± 0.2 | 34.0 ± 0.0 |
| MLDG [44] | 51.1 ± 0.3 | 14.1 ± 0.3 | 40.7 ± 0.3 | 11.7 ± 0.1 | 52.3 ± 0.3 | 42.7 ± 0.2 | 35.4 ± 0.0 |
| CORAL [70] | 51.2 ± 0.2 | 15.4 ± 0.2 | 42.0 ± 0.2 | 12.7 ± 0.1 | 52.0 ± 0.3 | 43.4 ± 0.0 | 36.1 ± 0.2 |
| MMD [46] | 16.6 ± 13.3 | 0.3 ± 0.0 | 12.8 ± 10.4 | 0.3 ± 0.0 | 17.1 ± 13.7 | 0.4 ± 0.0 | 7.9 ± 6.2 |
| DANN [24] | 45.0 ± 0.2 | 12.8 ± 0.2 | 36.0 ± 0.2 | 10.4 ± 0.3 | 46.7 ± 0.3 | 38.0 ± 0.3 | 31.5 ± 0.1 |
| CDANN [49] | 45.3 ± 0.2 | 12.6 ± 0.2 | 36.6 ± 0.2 | 10.3 ± 0.4 | 47.5 ± 0.1 | 38.9 ± 0.4 | 31.8 ± 0.2 |
| MTL [5] | 50.6 ± 0.2 | 14.0 ± 0.4 | 39.6 ± 0.3 | 12.0 ± 0.3 | 52.1 ± 0.1 | 41.5 ± 0.0 | 35.0 ± 0.0 |
| SagNet [54] | 51.0 ± 0.1 | 14.6 ± 0.1 | 40.2 ± 0.2 | 12.1 ± 0.2 | 51.5 ± 0.3 | 42.4 ± 0.1 | 35.3 ± 0.1 |
| ARM [87] | 43.0 ± 0.2 | 11.7 ± 0.2 | 34.6 ± 0.1 | 9.8 ± 0.4 | 43.2 ± 0.3 | 37.0 ± 0.3 | 29.9 ± 0.1 |
| VREx [42] | 39.2 ± 1.6 | 11.9 ± 0.4 | 31.2 ± 1.3 | 10.2 ± 0.4 | 41.5 ± 1.8 | 34.8 ± 0.8 | 28.1 ± 1.0 |
| RSC [36] | 39.5 ± 3.7 | 11.4 ± 0.8 | 30.5 ± 3.1 | 10.2 ± 0.8 | 41.0 ± 1.4 | 34.7 ± 2.6 | 27.9 ± 2.0 |
| Meta-DMoE [89] | 51.5 ± 0.9 | 15.4 ± 0.5 | 42.0 ± 0.6 | 11.9 ± 0.4 | 50.9 ± 0.2 | 44.0 ± 0.3 | 36.0 ± 0.2 |
| SelfReg [39] | 47.9 ± 0.3 | 15.1 ± 0.3 | 41.2 ± 0.2 | 11.7 ± 0.3 | 48.8 ± 0.0 | 43.8 ± 0.3 | 34.7 ± 0.2 |
| MixStyle [91] | 49.1 ± 0.4 | 13.4 ± 0.0 | 39.3 ± 0.0 | 11.4 ± 0.4 | 47.7 ± 0.3 | 42.7 ± 0.1 | 33.9 ± 0.1 |
| Fish [69] | 51.5 ± 0.3 | 14.5 ± 0.2 | 40.4 ± 0.3 | 11.7 ± 0.5 | 52.6 ± 0.2 | 42.1 ± 0.1 | 35.5 ± 0.0 |
| SD [60] | 51.3 ± 0.3 | 15.5 ± 0.1 | 41.5 ± 0.3 | 12.6 ± 0.2 | 52.9 ± 0.2 | 44.0 ± 0.4 | 36.3 ± 0.2 |
| CAD [65] | 45.4 ± 1.0 | 12.1 ± 0.5 | 34.9 ± 1.1 | 10.2 ± 0.6 | 45.1 ± 1.6 | 38.5 ± 0.6 | 31.0 ± 0.8 |
| CondCAD [65] | 46.1 ± 1.0 | 13.3 ± 0.4 | 36.1 ± 1.4 | 10.7 ± 0.2 | 46.8 ± 1.3 | 38.7 ± 0.7 | 31.9 ± 0.7 |
| Fishr [62] | 47.8 ± 0.7 | 14.6 ± 0.2 | 40.0 ± 0.3 | 11.9 ± 0.2 | 49.2 ± 0.7 | 41.7 ± 0.1 | 34.2 ± 0.3 |
| ITTA [15] | 50.7 ± 0.7 | 13.9 ± 0.4 | 39.4 ± 0.5 | 11.9 ± 0.2 | 50.2 ± 0.3 | 43.5 ± 0.1 | 34.9 ± 0.1 |
| ERM+ | 51.3 ± 0.1 | 15.8 ± 0.4 | 42.3 ± 0.1 | 13.0 ± 0.2 | 51.4 ± 0.4 | 44.3± 0.1 | 36.4 ± 0.1 |
| Ours | 50.7 ± 0.7 | 15.7 ± 0.0 | 41.5 ± 0.5 | 12.4 ± 0.2 | 51.4 ± 0.3 | 44.8± 0.2 | 36.1 ± 0.1 |

