# OpenReview forum: "LFME: A Simple Framework for Learning from Multiple Experts in Domain Generalization"
_NeurIPS.cc/2024/Conference — NeurIPS 2024 poster_

### Official Review · Reviewer_h8LM · 2024-07-09

**Soundness:** 3
**Presentation:** 3
**Contribution:** 3
**Rating:** 8
**Confidence:** 4

**Summary:**

This paper proposes a simple framework called LFME for learning from multiple experts in domain generalization. LFME introduces a logarithmic regularization term to enforce similarity between the target model and expert models, allowing the target model to acquire expertise from all source domains and perform well in any testing domain. Experimental results demonstrate that LFME outperforms existing techniques not only in classification and segmentation tasks but also reveals through in-depth analysis its ability to implicitly use more information for predictions and to discover difficult samples from experts, thereby enhancing generalization capability.

**Strengths:**

This paper proposes a novel idea on the basis of a new knowledge distillation paradigm for DG. The motivation of using multiple experts for improving generalization is clear and reasonable. Meanwhile, I also found the paper well-presented and easy to understand.

After checking the pseudocode and the provided code, I found the implementation extremely simple. Given the impressive results computed with rigorous evaluation protocols, I believe this paper can offer valuable contributions to the literature by providing an accessible and effective solution.

The provided deep analyses that explains why their method works are appreciated and, in my opinion, crucial for a paper, especially in the era of deep learning. Coupled with the empirical evidence in their supplementary material, I think most of claims made in their paper can be well supported.

**Weaknesses:**

Regarding the proposed method, although large improvements over the baseline ERM are observed, their method involves higher training costs. I noticed the authors provide the training time comparisons with different methods in their supplementary material, which shows that their LFME requires 1.5 times more training time than ERM, and it is also one of the most time-consuming methods compared. While I agree with the authors that requiring more training time is inevitable for methods designed based on KD, the authors should at least list this as a major drawback in their limitation section.

As in recent times, Vision Transformers (ViTs) have demonstrated substantial improvements in the classification and semantic segmentation tasks. Although the authors conducted experiments using different ResNet models, they should also consider adopting a ViT backbone for their experiments. This could provide valuable insights into the performance of their method when applied to more advanced architectures, potentially highlighting further benefits or limitations. Including such experiments would enhance the comprehensiveness of the study and align it with current trends in the field.

**Questions:**

In Section 4.2, the authors claim that their method improves DG by explicitly mining hard samples from the experts. However, Tables 3 and 6 show that specifically mining hard samples from the experts improves performance of ERM only in the TerraInc dataset, but not in the PACS dataset (with an average improvement of only 0.1 pp). Could the authors provide an explanation for this phenomenon?

**Limitations:**

Authors have adequately addressed the limitations.

---

> ### Author Rebuttal · Authors · 2024-08-06
>
> > W1: List more computational cost as a limitation.
>
> A1: We thank the reviewer for the suggestion, we will include this limitation in our revised paper.
>
> > W2: Experiments with ViTs.
>
> A2: We conducted further experiments by evaluating our method, the baseline ERM, and some leading methods in Tab. 1 with a ViT-Base model using the same setting detailed in the manuscript. Results in Tab. 7 in the general response show that our method can still obtain favorable performance against existing arts despite using different networks.
>
> > Q1: Why hard sample mining can improve performance in TerraInc dataset but not the PACS dataset.
>
> A1: We infer this is mainly because compared to the PACS dataset, TerraInc is more difficult for the corresponding expert, which can provide more hard samples to boost generalization (briefly mentioned in Line 623 in our paper). As shown in Tab. 5 in the manuscript, the experts can achieve an average accuracy of 96.7% when tested with the source data in the PACS dataset, indicating that there are very few samples in PACS that can be regarded as hard for the experts. On the other hand, the in-domain results for the experts in TerraInc are 91.1% (Tab. 1 in the general response), suggesting that the dataset can provide more hard samples to guide the target model than PACS, thus improving the performance. We will include this analysis in our revised paper.

---

> ### Comment · Reviewer_h8LM · 2024-08-12
>
> The author's rebuttal allayed some of my concerns, and I chose to raise my rating, considering that the paper still had some sparkle to it.

---

### Official Review · Reviewer_uZyo · 2024-07-11

**Soundness:** 2
**Presentation:** 2
**Contribution:** 2
**Rating:** 4
**Confidence:** 3

**Summary:**

This paper focuses on improving domain generalization by utilizing multiple experts. Particularly, a simple yet effective framework is proposed whereby a target (student) model is learned from multiple expert (teacher) models through logit regularization. After learning, the target model can grasp knowledge from multiple source domains and shows advantages in handling hard samples. The proposed approach demonstrates consistent improvements across different evaluation tasks, outperforming existing SOTAs.

**Strengths:**

This paper overall is well written. It provides in-depth theoretical insights on its proposed logit regularization and surrogate ablation studies.

**Weaknesses:**

1: The contribution of the proposed approach is marginal. The performance of the proposed approach does not show much improvement compared to ERM or expert models (Table 5).

2: The baseline models for comparison are not clearly introduced, which creates difficulties in understanding the Tables.

3: The comparison to the approaches that directly learn a foundation model using data from all source domains is not discussed. Hence, the benefits of the proposed approach, which have to use a set of expert models plus a target model, compared to the foundation model are not clear.

**Questions:**

1: What is the difference between the proposed approach and foundation models? Why not directly learn a model using all the source domains? How does the performance of the proposed approach compare to Segment Everything?

2: What’s the difference between the proposed approach and the baseline ERM? In lines 77-78, the authors mention ‘only one hyper-parameter’ without further illustration. The difference is clarified only until line 138. Statements near lines 77-78 need to be improved to avoid confusion. More importantly, it can be misleading to say “only one additional parameter upon ERM” since a target model is also involved.

3: How does the value q_*^E relate to the hardness of samples? What causes the sample to be ‘hard’? Low data frequence or image quality?  Without these clarified, the provided theoretical insights are not convincing.

**Limitations:**

No discussions on limitations and potential negative societal impact of the work are provided.

One limitation is that the performance of the proposed approach can be upper bounded by the performance of the expert models. This limitation again raises the question of why not directly learn from all source domains?

---

> ### Author Rebuttal · Authors · 2024-08-06
>
> > W1: The performance does not show much improvements in Tab. 5
>
> A1: We want to clarify that the in-domain comparisons in Tab. 5 are only used to verify our claim that the target model has evolved to be an expert in all source domains. In the DG setting, out-of-domain performance is often regarded as the metric for evaluating different methods, and our method is shown to outperform many sophisticated designs. When compared to ERM and the different expert models, our method also shows large improvements (Tab. 4). We note that the out-of-domain performance of our method is regarded as impressive by other reviewers.
>
>
> > W2: The compared baseline models are not clearly introduced.
>
> A2: The primary baseline models (in both Tab. 1 and 2) are ERM that directly trains the model with data from all source domains. According to the existing benchmark [20], in the DG task, ERM is a strong baseline that can outperform many SOTAs when evaluated with strict evaluation protocols. As for other compared models, we use most of the implementations in the commonly-used DomainBed [20] for evaluation. It would be better to refer to the benchmark for details.
>
>
> > W3: Comparisons with foundation model that uses data from all source domains is not discussed.
>
> A3: We want to clarify that the referred foundation model is ERM, and we include comparisons with it in all our experiments (i.e. Baseline in Tab. 2 and ERM in all other tables), where our design can improve the performance in almost all evaluated datasets.
>
> > Q1 (part 1): Difference between the proposed approach and foundation models, why not directly learn a model using all the source domains?
>
> A1 (part1): Compared to the foundation model, the target model is trained with an additional logit regularization term $\mathcal{L}_{guid}$ (without it, the target model degrades to the foundation model), which is designed to help the target model to be expert in all source domains. As explained in Sec. 4, we reveal that the logit regularization term can help the target model to use more information and mine hard samples from the experts, which are both beneficial for improving generalization.
>
> > Q1 (part 2): Why not compare with the Segment Anything Model (SAM).
>
> A1 (part2): Our experiment settings in the generalizable semantic segmentation task are the same as previous mainstreams [32, 48, 12], which train the target model on synthetic data and test it in unseen real-world scenes. We do not compare with SAM mainly because (1) SAM and mainstreams use different network structures: SAM uses ViT and CLIP text encoder in their model, which has a total of 636M parameters. Differently, the mainstreams use DeepLabv3+ ResNet50 as the backbone, consisting of 39M parameters, which is only 6 percent the size of SAM; (2) SAM is trained over 1 billion data, while the synthetic data used for training in our setting is only 34,366. We thus do not compare with SAM.
>
> Our work focus the generalization task, which can be applied to many downstream vision tasks, semantic segmentation included. Unlike SAM that aims to solve segmentation with large models and more data, this work can be applied to situations when there is only limited training data from certain domains, and the task is to generalize the model to new domains, for example in particular medical situations. We leave such exploration to potential future works.
>
> > Q2 (part 1): What’s the difference between the proposed approach and the baseline ERM.
>
> A2 (part 1): ERM is the referred foundation model that is trained with all data (by using only $\mathcal{L}_{cla}$ in Eq. (3)). Please refer to our response for Q1 (part 1) for their differences.
>
> > Q2 (part 2): Improper to say “only one additional parameter upon ERM” since a target model is involved.
>
> A2 (part2): We appreciate the suggestion. This sentence is to underline that except for the only one weight parameter, our model does not involve more heuristic hyper-parameter settings compared to ERM. We will clarify it in our revised version to avoid confusion.
>
> > Q3: How does $q_{\ast}^E$ relates to hard samples and what are those hard samples?
>
> A3: As the classification loss is computed as $-\sum y \log q$, smaller $q$ indicates larger loss, which corresponds to a more difficult sample. This is is consistent with the previous work [27]. In Sec. D.4 in the appendix, we find that hard samples from the domain experts contain more domain ambigous data than that from the model fed with all data, explaning why domain experts must be involved. Please also refer to our general response for details.
>
> > L1: No discussions on limitations and potential negative societal impact of the work are provided.
>
> A1: Please note that we discuss the limitation of this work in Sec. A in the appendix. For the potential negative societal impact, this work designs a new idea for the task of domain generalization that holds significant potential for a wide array of real-world applications. It may bear potential societal consequences similar to other machine learning systems. We will include this in our revised paper.
>
> > L2: The performance of the proposed approach can be upper bounded by the performance of the expert models, why not directly learn from all source domains?
>
> A2: Our method outperforms domain experts in both out-of-domain (i.e., Tab. 4) and in-domain (i.e., Tab. 5) settings, indicating that the performance is not bounded by domain experts. This is because the target model uses outputs from all experts and gt as training labels (see Eq. (3)), thus the experts can not bound the performance of the target model. When compared to the referred foundation model that directly trains with all data, our method also shows large improvement (i.e., Tab. 1 and 2). Please refer to our response for Q1 (part 1) for the explanations.

---

### Official Review · Reviewer_xxxE · 2024-07-12

**Soundness:** 4
**Presentation:** 4
**Contribution:** 4
**Rating:** 8
**Confidence:** 5

**Summary:**

This paper addresses the problem of domain generalization from the perspective of  ''learning multiple experts''. In particular, they propose to train multiple experts specialized in different domains, whose output probabilities provide professional guidance by simply regularizing the logit of the target model. The proposed logit regularization provides effects of enabling the target model to harness more information, and mining hard samples from the experts during training. Experiments on standard DG benchmarks demonstrate the effectiveness of the proposed framework.

**Strengths:**

- The overall manuscript is well-organized and easy to follow.

- The method is simple and effective. Its impressive results in different tasks demonstrate the strong applicability of this idea to potential other tasks.

- The related work section is comprehensive, especially regarding comparisons with Meta-DMoE that also tries to distill knowledge from domain experts.

- The idea of using MSE loss between logit and probability for distillation is new and well-explained, supported by both theoretical and empirical evidence.

- The free lunch idea introduced in sec. 6.1 is interesting and seems to have the potential for more general applications.

**Weaknesses:**

While adopting the idea of knowledge distillation (KD) and the concept of domain experts for improving DG is not new in the literature, it is appreciated that this method can be implemented in such simple form and be explained with extensive insights. Nevertheless, there are still a certain amount of claims in the paper that requires proper explanations or more justification:

- Performance reported in the original Meta-DMoE paper is higher than that in Table 1. For example, Meta-DMoE reports an average of 86.9 in PACS, surpassing the best result listed in the table. The authors should provide an explanation for this discrepancy. Moreover, I noticed that the training time comparisons between these two methods are provided, it would also be beneficial to highlight the test and training resource differences between these two methods in the related work section.

- The authors mention that their method introduces only one hyper-parameter, which is within a certain range. It would be beneficial to include an ablation study to explain why this specific range was chosen.

- It would be beneficial to evaluate this method with more real-world DG problem, such as using dataset from wilds benchmark.

- The method [1] with similar settings should be compared in Table 1. Additionally, some methods, such as MIRO, report improved performance when combined with SWAD. Including these results in Table 1 would provide a more comprehensive comparison.

- Minor issues. Formats of ”i.e.” in line 145, ”Tab.” in line 578, and ”Fig.” in line 581 are inconsistent with their usages in other places. The authors should standardize the format of similar abbreviations to maintain uniformity.

Reference:

[1] Domain Generalization via Rationale Invariance, in ICCV 2023.

**Questions:**

- This method needs to train additional expert per domain, what if there are large amount of domains for the training data, any potential solution for this occasion?

- Since multiple expert models are required when training the target model, how would this method be applied to the training process of large foundation models?

- The free lunch idea appears to be applicable beyond just the DG setting. How does it perform on ImageNet? Including such experiments could enhance the significance of this concept.

**Limitations:**

yes

---

> ### Author Rebuttal · Authors · 2024-08-06
>
> > W1: Performance in the Meta-DMoE paper is higher than that in Tab. 1, and highlight the resource usage differences.
>
> A1: Note that we mainly use the ResNet18 model for evaluating Meta-DMoE in Tab. 1, and the experiments are conducted for a total of 3x20 trials using the default hyper-parameter settings that are within large ranges. These settings are different from those in the original paper (where ResNet50 results are reported), which can cause the differences in the reported numbers of Meta-DMoE. We will include this clarification in our revised paper.
>
> We thank the reviewer for the suggestion. We list the efficiency comparisons in our response regarding w1 for Reviewer oULp, and we will include the differences in the related work section.
>
> > W2: Conducting ablation study to analyse the selected value range for the weight parameter.
>
> A2: We conduct this ablation study in Tab. 4 in our general response, which shows that our method can obtain relatively better performance when the weight parameter is selected wihtin [0.1, 10].
>
> > W3: Experiments in datasets from the Wilds benchmark.
>
> A3: We conduct experiments using three real-world datasets (i.e., iWildCam, RxRx1, and FMoW) using the default settings in Wilds. Results listed in Tab. 5 in the general response show that our method can improve the baseline in all three datasets and obtains favorable performance against existing arts.
>
> > W4: Including RIDG [1] and MIRO+SWAD in Tab. 1.
>
> A4: We include comparisons with the referred methods in Tab. 6 in our general response. The comparisons will be included in Tab. 1 in our revised paper.
>
> > W5: Minor issues.
>
> A5: We thank the reviewer for the notifications, these typos will be corrected in our revised paper.
>
> > Q1: Solution for datasets with a large number of domains.
>
> A1: When encountering datasets containing numerous domains, we can utilize the existing strategy [a] that clusters the training domains into fewer super-domains, thus reducing the expert number. We leave this exploration to potential future works.
>
> [a] Orbit: A real-world few-shot dataset for teachable object recognition, in ICCV'21
>
> > Q2: How to apply the method to the training process of large foundation models?
>
> A2: When applied to the training process of large foundation models, a possible solution for easing the memory constraint problem is to use a sequential training strategy that first learns different experts, and then trains the target model without optimizing these experts. In this way, professional guidance from different experts can still be obtained, and we can avoid the memory constraint for training $M+1$ models at the same time (with $M$ the expert number).
>
> > Q3: Including experiments of the free lunch idea in Imagenet setting.
>
> A3: We conduct experiments with the free lunch idea in Imagenet as suggested. Compared to the baseline, the free lunch can obtain better performance by 0.4pp for the top1 acc (76.9 vs. 76.5) and 0.5pp for the top5 acc (93.7 vs. 93.2). These results show that the free lunch idea is beneficial for large datasets.

---

> > ### Comment · Reviewer_xxxE · 2024-08-11
> >
> > Thank you for the detailed responses. My previous concerns have been well addressed. After carefully reviewing the other reviewers' comments and the authors' replies, I believe the paper has no significant flaws, and therefore, I choose to maintain my score.

---

### Official Review · Reviewer_oULp · 2024-07-12

**Soundness:** 2
**Presentation:** 2
**Contribution:** 2
**Rating:** 4
**Confidence:** 4

**Summary:**

The computer-vision paper introduces a strategy of learning from multiple experts (LFME), which performs knowledge distillation from models specially trained on data from different domains. In particular, the experts are trained jointly with the target model, and a specific form of logit regularization is chosen for knowledge distillation due to empirical verification. The idea is relatively straightforward to understand with strong empirical results on image classification (by Domainbed) and semantic segmentation.

**Strengths:**

1. The experimental results come in strong when compared to many other baselines. Moreover, the study has extended results for semantic segmentation.
2. The visualizations are helpful to illustrate the ideas and evidence.
3. The empirical study on naive aggregation of domain experts shows that naive ensemble does not necessarily lead to better generalization. Then ablation study shows the chosen form of logit regularization outperforms other knowledge distillation alternatives.

**Weaknesses:**

1. The novelty is relatively limited. The proposed approach is very similar to Meta-DMoE with a few changes such as logit regularization and alternating updates, but these changes are not well justified (see below). The idea of utilizing multiple domains have also been explored by [1].
2. There is a lack of coherent argument to understand the proposed approach. In Section 4.1, the paper briefly discussed how the additional logit regularization enables learning more information. In addition, in Section 6.2, the proposed approach LFME is compared with Label Smoothing. However, the discussion is not very convincing. From the perspective of using information from other classes, I do not see the fundamental difference between logit regularization and Label Smoothing (LS). Both logit regularization and LS have hyperparameters to be tuned, so it might not be appropriate to claim its advantage as "not involve hand-crafted settings” (line 187) and criticize LS has “potential improper heuristic designs” (189).
3. Moreover, in Section 4.2, the paper justifies the advantage of the proposed approach LFME by “mining hard samples from the experts”. It is a general statement that is true for low-confidence predictions from any model, so it does not constitute a strong argument. Intuitively, the in-domain samples are easier than out-of-domain samples for each expert, and the hard samples aggregated from all experts are not specifically representative of any subpopulations. The argument will be more comprehensive if the paper can show what are these hard samples and why it matters to mine the hard samples from the domain experts (and not an ensemble of experts or some random experts).
4. In Section 6.5, the result for in-domain evaluation is only provided for one dataset. This is not sufficient to justify a strong claim that the target model is an expert on all source domains.
5. Code is submitted but it is unclear how the proposed approach can be combined with SWAD. Does it apply to just the target model or also the domain experts? The hyperparameter configurations to obtain the SoTA results are also unclear for reproducibility.

[1] Yao, Huaxiu, Xinyu Yang, Xinyi Pan, Shengchao Liu, Pang Wei Koh, and Chelsea Finn. "Leveraging domain relations for domain generalization." *arXiv preprint arXiv:2302.02609* (2023).

**Questions:**

1. In Algorithm 1, the target model and the experts are trained simultaneously, which is a more parallel and iterative scheme rather than a sequential one with two stages, i.e., training the experts first, then distilling the knowledge to the target model using logit regularization. Why is such a design choice made? Are there better justifications than just “for simplicity”? Please explain with more details.
2. Each expert is generally weak as they are only trained using the data from their own domain. Is it necessary and how to ensure the quality of the experts?
3. In Appendix A, Limitations, the authors discussed that LFME cannot be applied to single-domain tasks. How does the number of domains affect the proposed approach? Will the algorithm benefit from having more domains with less data or more data in fewer domains?

**Limitations:**

The limitations has been discussed in Appendix A.

---

> ### Author Rebuttal · Authors · 2024-08-06
>
> > W1. Novelty
>
> A1: Differences between Meta-DMoE and LFME are as follow.
>
> 1. ### Domain experts in these two works serve different purposes
>
> Meta-DMoE aims to adapt the trained target model to a new domain in test. To facilitate adaptation, their target model should be capable of identifying domain-specific information (DSI). Therefore, the target model is enforced to extract DSI similar to those from domain experts. Notably, their trained experts are expected to thrive in all domains, which are used to extract DSI **not** in their trained domains but rather in an unseen one. In a word, domain experts in Meta-DMoE serve as supervision to ensure that the target model can adapt to a new domain. Differently, LFME expects its target model to be expert in all source domains. In our framework, domain experts provide professional guidance for the target model only in their corresponding domains.
>
> 2. ### Different implementations
>
> Meta-DMoE involves meta-learning and test-time training (TTT), where domain experts are used for data from their unseen domains in both training and test. The overall process can be time-consuming due to TTT and the second-order optimization in meta-learning, which may pose efficiency problems compared to the straightforward training in LFME. Meanwhile, their idea is developed on the traditional KD idea that enforces feature similarity between teachers and student, which is shown suboptimal in our analysis.
>
> 3. ### Different effectiveness and efficiencies
>
> When evaluated with the same strict setting in DomainBed, LFME leads Meta-DMoE in all datasets with an average improvement of 1.8pp. The required running time for LFME is also less than Meta-DMoE: 38' vs. 73' for one trial in PACS (even without the experts training time, Meta-DMoE still requires 45' for one trail).
>
> We list comparisons with Meta-DMoE in Sec. 2, and are glad that Reviewer xxxE find it comprehensive. Regarding the idea in [1], it shares similar design with DAELDG that uses different classifiers as experts and averages the final outputs of all experts as the result. Similar intuition has been discussed in our analysis. We will discuss this work in the revised version.
>
> > W2: More justifications for argument in Sec. 4.1 and comparisons against LS
>
> A2: We provide more justifications (visual examples included) for further comprehending the analysis in Sec. D.1. More empirical evidence supporting this argument is provided in D.2. Please refer to them for a better understanding.
>
> LS uses a heuristic $\epsilon$ to adjust output probability and prevent overconfidence in the gt label: $(1-\epsilon)H(q, y) + \epsilon H(q, U)$ (given U a uniform distribution). if $\epsilon$ is improperly set (e.g. close to 1) and the dominant label approximates $U$, LS may fail. Differently, LFME uses labels from both gt and experts' outputs to excel in all source domains: $H(q, y) + \frac{\alpha}{2}||z-q^E||^2$, where we do not need to deliberately calibrate the probability. Those two methods differ inherently. Resulsts in Sec. 6.2 also show LFME performs better than latest LS methods. Notably, experiments in Tab. 4 in the general response show that the model is insensitive w.r.t $\alpha$ as it is on par with ERM even with $\frac{\alpha}{2}=1000$, because $q_{\ast}^E$ is mostly aligned with $y$ (see Fig. 2 (a)), which is a much preciser label than $U$. We thus say that LS bears a con of potential improper heuristic designs against LFME. Meanwhile, LS treats all samples equally, while LFME can specifically focus more on hard samples. That is another pro for LFME.
>
> > W3: What are those hard samples and why use domain experts
>
> A3: The hard samples are domain ambigous data, aligned with your intuition. We find that domain experts can locate more of these data than random models, thus can better help DG. Please see our general response for details.
>
> > W4: More in-domain evaluations
>
> A4: We conduct experiments on other two datasets (i.e. TerraInc and VLCS) and list the results in Tab. 1 in the geneal response. Results are consistent with that in Tab. 5, where the target model can lead the baseline and experts in all cases, validating that it becomes expert on all source domains.
>
> > W5: SWAD and hyper-parameter (HP) settings
>
> A5: We apply SWAD only for the target model. We use the default settings in original implementations of Domainbed and SWAD. For your reference, we list all HP in Tab. 2 in the general response.
>
> > Q1: Why use parallel training
>
> A1: In the parallel scheme, each training sample will go through 2 forward passes (one for expert and another for the target), and we can use a single optimization to update all parameters and a total of 5000 updating steps for a trial in PACS (see Alg. 1). Instead, if using the sequential scheme, each sample will go through 3 forward passes (one for training expert, and two for training the target), and we will require 2x5000 updating steps for a trial. Training times for these two schemes are 38' (parallel) vs. 55' (sequential), we thus use parallel scheme for simiplicity.
>
> > Q2: Is it necessary to ensure the experts' quality
>
> A2: Different from Meta-DMoE, experts in LFME are expected to saturate only on their corresponding domains. Although less data is used, they can still outperform the baseline within their trained domains (see our in-domain experiments). Given that our interest lies primarily in the outputs of these experts within their designated training data, their general proficiency beyond these specific data is of secondary concern. Thus, it may be unnecessary to ensure the quality of the experts across broader domains.
>
> > Q3: How does the number of domains affect the proposed approach, will the method benefit from more domains or more data
>
> A3: We answer this question by merging the source domains to create varying domain numbers. As shown in Tab. 3 in the general response, the model performance is generally positively correlated w.r.t domain numbers.

---

> > ### Comment · Reviewer_oULp · 2024-08-09
> > **Reviewer Response**
> >
> > Thank you for the detailed responses and comprehensive experiments, which have addressed most of my concerns.
> >
> > However, I am still not convinced by the arguments on mining the hard samples and the analysis of why the proposed approach works. In Sec. 4.2, the rescaling factor $\mathcal{F}, \mathcal{F}’$ are shown to be negative in Figure 2(e). While it is true that the magnitude is larger for harder examples, if the gradient direction is reversed, then the model is no longer learning the sample as it's going the opposite of the original ERM gradient. I am not sure if we can still interpret this as "mining" the hard samples because they are somewhat being "forgotten". Please clarify.

---

> > > ### Author Response · Authors · 2024-08-09
> > > **Thank you for the feedback**
> > >
> > > We sincerely thank the reviewer for the prompt reply!
> > >
> > > We would like to clalrify that if two networks have different gradient directions, it does not mean one network is "forgetting" the samples compared to the other. The negative values of $\mathcal{F}$ and $\mathcal{F}'$ apply for all training samples, but this does not imply that LFME is forgetting all samples in contrast to ERM. Since hard samples from experts correspond to larger gradient magtitude for the target model, they will have more significant impacts on the target model's updating, which can shift the decision boundaries not aligned with easy samples that with small gradient magtitude, potentially causing these easy samples to be overlooked by the target instead. We thus say the target model can mine hard samples from the experts.
> > >
> > > The argument in Sec. 4.2 can be better interpreted from the following example: assuming two samples $x_1$ and $x_2$ from the same class and domain, with the target model producing identical output logits for both: $z_1 = z_2$. If the corresponding expert has different levels of confidence regarding these samples, such that $q_{\ast 1}^{E} > q_{\ast 2}^{E}$, the analysis in Sec. 4.2 suggests that $x_2$ will have larger impacts on the updating of the target model than $x_1$, and the target model will focus more on $x_2$.
> > >
> > > We thank again for the reviewer's time, and wish our clarification is helpful.

---

> > > > ### Comment · Reviewer_oULp · 2024-08-12
> > > > **Reviewer Response**
> > > >
> > > > Practically as the algorithm is working, I would like to believe that LFME is not forgetting all samples. However, the math and plot in Section 4.2 say otherwise.
> > > >
> > > > This is because originally we are doing gradient descent using ERM. When we multiply a negative value in front of it, it becomes gradient ascent instead. According to the plot, it's happening across most of the iteration steps, which is quite puzzling. I am not sure how is $\mathcal{F}$ calculated practically, but it might be helpful to clarify it.
> > > >
> > > > Moreover, if we just look at Eq. (4), if $q_{*1}^{E} > q_{*2}^{E}$, the direct interpretation is to reduce the logit value of $x_2$ more than $x_1$, so that the model should not be overconfident about it. I honestly couldn't see how this is mining the "hard" sample from an "unconfident" expert.

---

> ### Author Response · Authors · 2024-08-12
> **Response to further concerns**
>
> We appreciate the reviewer's feedback and thoughtful response. We answer the additional questions below.
>
> > Q: This is because originally we are doing gradient descent using ERM. When we multiply a negative value in front of it, it becomes gradient ascent instead.
>
> A: We want to clarify that our only objetive is to minimize the loss $\mathcal{L}_{all}$ in Eq. (3) throughout the training process. It's important to emphasize that there is no gradient reversal operation in our framework; specifically, we are not performing gradient descent using ERM followed by multiplying a negative value for gradient ascent.
>
> > Q: How is $\mathcal{F}$ calculated.
>
> A: we want to clarify that we do not need to compute $\mathcal{F}$ during updating, $\mathcal{F}$ is used as an analytical tool to show how much the gradient magtitudes is rescaled w.r.t to the baseline model that without using the logit regularization term, by comparing ratio of gradients from the two models [64]. In our case, gradients of the two models (LFME and ERM) are $q_c - y_c + \alpha (z_c - q_c^E)$ (e.g. $\frac{\partial \mathcal{L}\_{all}}{\partial z_c}$) and $q_c - y_c$ (e.g. $\frac{\partial \mathcal{L}\_{cla}}{ \partial z_c}$), there gradient ratio is thus $\frac{\partial \mathcal{L}\_{all}}{\partial z_c} / \frac{\partial \mathcal{L}\_{cla}}{ \partial z_c}$, which leads to the computing of $\mathcal{F}$ and $\mathcal{F}'$ in Eq. (6) and (8).
>
> > Q: In the given example, how the mining the "hard" sample from an expert achieved and its relation to the reduction of the logit value.
>
> A: For the given example, where $z_1 = z_2$, it indicates that the model is with equal confidence w.r.t the two samples. Without experts involvements, these two samples will contribute equally to the target model's updating. However, when the expert is involved and $q_{\ast 1}^{E} > q_{\ast 2}^{E}$，Eq. (6) and (8) indicate $|\mathcal{F}_1| < |\mathcal{F}_2|$ and $|\mathcal{F}'_1| < |\mathcal{F}'_2|$: the gradient magtitude from $x_2$ is changing more significantly than that from $x_1$ both on the basis ERM, allowing $x_2$ to have a greater impact on the target model's update. The derivations imply that the expert can expose its hard samples to the target model, which is not applicable in ERM.
>
> From your point of view, the objetive in Eq. (3) implicitly encourages the target model to be less confident w.r.t $x_2$. We want to clarify that this confidence reduction does not conflict with mining $x_2$, because lower confidence results in a higher training loss, prompting the target model to explore the corresponding sample more thoroughly, where both the dominant features for classification and other information shared with other classes will be explored, with the dominant features being the primary focus (because both $\mathcal{L}\_{cla}$ and $\mathcal{L}\_{guid}$ in $\mathcal{L}\_{all}$ encourage the gt logit to be the largest, which also indicates that lower confidence in LFME does not equate to lower accuracy—a fact validated by our in-domain results). On the contrary, high confidence indicates that the model may overlook the sample after obtaining its dominant features, making it less important on the updating process. Moreover, penalizing high-confidence outputs has been shown to be effective in improving a variety of learning tasks [a], our objective can thus be viewed as specifically mining hard samples from the experts for penalization. From this perspective, in our case, penalizing confident can be integrated within our explanation of hard sample mining.
>
> We sincerely thank you for your invaluable time and the fruitful discussions. We are more than willing to provide any additional justifications or clarifications you may need.
>
> [a] Regularizing Neural Networks by Penalizing Confident Output Distributions, in ICLR'17

---

> > ### Comment · Reviewer_oULp · 2024-08-12
> > **Reviewer Response**
> >
> > Thank you for the prompt clarifications.
> >
> > Unfortunately, I am still not convinced and couldn't agree with the counterintuitive rationale.
> > > Specifically, we are not performing gradient descent using ERM followed by multiplying a negative value for gradient ascent.
> >
> > I understand LFME is not doing this explicitly, but aren't they mathematically equivalent? If it doesn't align, then how does the analysis in Section 4.2 provide valid justifications? I think when we are trying to investigate the working mechanism of the proposed algorithm, the story needs to be coherent and not mislead the readers, especially when the analysis consists of a major section of the main paper.
> >
> > > Penalizing high-confidence outputs has been shown to be effective in improving a variety of learning tasks
> >
> > As far as I understand, arguing from the confidence perspective might be more reasonable than hard sample mining.

---

> ### Author Response · Authors · 2024-08-12
> **Reply to your concerns**
>
> We thank the reviewer for the prompt reply.
>
> > Q: Is updating LFME mathematically equivalent to doing gradient descent using ERM followed by multiplying a negative value for gradient ascent?
>
> A: We want to clarify that the two gradient parts of $\mathcal{L}\_{all}$ w.r.t $z$ (e.g. $\partial \mathcal{L}\_{cla}$ and $\partial \mathcal{L}\_{guid}$) may better be interpreted as a whole in a updating step, as one updating step requires the combination of the two gradients for a joint classification task.
>
> According to your suggestion, if we specifically interpret the updating process by separately considering $\partial \mathcal{L}\_{cla}$ and $\partial \mathcal{L}\_{guid}$, we can see that in both terms, $z\_{\ast}$ correspond to largest labels. That is, $z\_{\ast}$ is encouraged to be close to $y\_{\ast}$ by minimizing $\mathcal{L}\_{cla}$ and close to $q\_{\ast}^E$ by minimizing $\mathcal{L}\_{guid}$. This means the updating process will maintain its discriminative ability, and the model will not forget the sample. The different gradient directions can be interpreted as $\mathcal{L}\_{cla}$ encourages $z\_{\ast}$ to become infinitely large, focusing on dominant features, while $\mathcal{L}\_{guid}$ optimizes $z\_{\ast}$ to the opposite direction to be close to $q\_{\ast}^E$ so that other information can also be involved. Therefore, $\partial \mathcal{L}\_{guid}$ should not be sees as tries to "unlearn" what has been learned by$\partial \mathcal{L}\_{cla}$. Rather, on the basis of not compromizing the target model's discriminative ability, they serve different purposes to encourage the model to focus on different types of features.
>
> > Q: Is mining hard samples from experts counterintuitive?
>
> A: The primary effect of knowledge distillation (KD)-based methods can be attributed to sample reweighting [b], where the weight of a training sample in the student model is related to the teacher's confidence. Although a different KD scheme is used, we apply the same rescaling analytical tool to demonstrate that one of the working mechanisms of the proposed KD form can still be seen as weighting different samples based on the teacher's confidence. Since different KD forms are utilized, the weighting strategies differ between the proposed LFME and the traditional KD approach where easier samples from teachers correspond to larger weights (comparisons between LFME and traditional KD ideas are presented in Sec. 6.3, where LFME performs better for the task of DG). Meanwhile, since hard samples from domain experts are more likely to be domain ambiguous data, enforcing the target model to learn these data more throughly can help it to explore more domain-invariant features, which are crucial for generalization [17]. In this regard, mining hard samples from domain experts is an intuitive motivation for improving DG.
>
> > Q: Arguing from the the confidence perspective.
>
> A: We thank the reviewer for the thoughtful suggestion. We want to clarify that the hard sample mining is aligned with the proposed confidence reduction perspective: in the given example, the target model will reduce confidence of $x_2$ more than that of $x_1$ even if they are with the same output logits, consistent with our explanation that hard samples from the experts will affect more on the target model's updating. Therefore, there may be no intrinsic difference between the hard sample mining and the suggested confidence reduction idea. Meanwhile, we provide empirical evidence in Sec. D.3 to show that the classification results of hard samples from corresponding experts are indeed improved in LFME, we thus argue the working mechanism from the hard sample mining perspective.
>
> On the other hand, merely analyzing the confidence is similar to our initial analysis in Section 4.1, where the idea of smoothing the output probability of the target model is equivalent to making it less overconfident.
>
> We sincerely thank you for your invaluable time and all thoughtful suggestions, which definetly help improve our paper. We wish these clarifications are helpful for easing your concerns.
>
> [b] Born-Again Neural Networks, in ICML'18.

---

### Author Rebuttal · Authors · 2024-08-06

We thank all reviewers for their hard work and insightful suggestions. We are inspired that Reviewer xxxE and h8LM find our work simple and Reviewer xxxE, h8LM, and oULp think the performance is strong. We are also glad that our in-depth theoretical insights are appreciated by Reviewer xxxE, h8LM, and uZyo.

For the general question raised by Reviewer oULp and uZyo, we answer them below:

>Q: How does $q_{\ast}^E$ relates to hard samples, what are those hard samples, and why mine hard samples from the domain experts?

A: A smaller $q_{\ast}^E$ is associated with larger cross-entropy loss for the corresponding expert, indicating that the sample is more difficult for the expert than others with smaller loss (i.e. larger $q_{\ast}^E$). This is consistent with the previous work that specifically aims to mine hard samples [27].

We conduct analyses in D.4 to study what are those hard samples and the differences between hard samples from domain experts and other models. Specifically, compared to hard samples from the model that trains with all data, we find that hard samples from domain experts contain more ambiguous data that locates in the mixed region or the boundary of different domains (shown in Fig. 6), indicating that they may encode more out-of-domain information. This is consistent with the assumption from Reviewer oULp.

By conducting experiments (results are listed in Tab. 6), we find that hard samples from domain experts is more beneficial for DG than hard samples from the model fed with all data. This is because emphasizing these domain ambiguous data can assist the target model in exploring more domain-invariant features that is consistent across different domains, which is crucial for improving model robustness [17]. This experiment justifies our design by involving domain experts (instead of some random models) in our framework. Please see D.4 for more details.

We conduct the following experiments and list the results in the uploaded file:

1. More in-domain evaluations from different models (Tab. 1);
2. Hyper-parameters used for reproducing results in the DomainBed benchmark (Tab. 2);
3. Effect of the number of domains on model performance (Tab. 3);
4. Sensitivity analysis regarding the weight parameter $\alpha$ (Tab. 4);
5. Evaluation results in datasets from the Wilds benchmark (Tab. 5);
6. Comparisons with more methods in DomainBed (Tab. 6);
7. Evaluating our method with vision transformer (Tab. 7).

All experiments in the rebuttal are conducted with the same experimental settings introduced in our paper where the DomainBed benchmark is utilized. These evaluations will be included in our revised manuscript.

We answer questions raised by each reviewer in the following.

---

### Decision · Program_Chairs · 2024-09-25

**Decision:**

Accept (poster)

**Comment:**

The paper proposes a framework that leverages multiple expert models to enhance domain generalization through logit regularization. While reviewers commended the clarity of the writing and the empirical results showing improvements over baselines, several concerns were noted: the approach's novelty is limited, there is a lack of sufficient justification for key methodological choices, and the evaluation does not cover a diverse range of tasks and datasets. the authors' rebuttal provided additional experiments and clarifications and addressed most of the concerns. Therefore, the recommendation is to accept the paper.